# Ultra-fast charging in aluminum-ion batteries: electric double layers on active anode

Xuejing Shen [1,2,5], Tao Sun[1,2,5], Lei Yang [1,3], Alexey Krasnoslobodtsev[3,4], Renat Sabirianov[3,4], Michael Sealy[2,4], Wai-Ning Mei[3,4], Zhanjun Wu [1✉] & Li Tan [2,4✉]

With the rapid iteration of portable electronics and electric vehicles, developing high-capacity batteries with ultra-fast charging capability has become a holy grail. Here we report rechargeable aluminum-ion batteries capable of reaching a high specific capacity of 200 mAh $g^{-1}$. When liquid metal is further used to lower the energy barrier from the anode, fastest charging rate of $10^4$ C (duration of 0.35 s to reach a full capacity) and 500% more specific capacity under high-rate conditions are achieved. Phase boundaries from the active anode are believed to encourage a high-flux charge transfer through the electric double layers. As a result, cationic layers inside the electric double layers responded with a swift change in molecular conformation, but anionic layers adopted a polymer-like configuration to facilitate the change in composition.

[1] School of Aerospace, Dalian University of Technology, Dalian 116024, China. [2] Department of Mechanical & Materials Engineering, University of Nebraska, Lincoln, NE 68588, USA. [3] Department of Physics, University of Nebraska, Omaha, NE 68182, USA. [4] Nebraska Center for Materials and Nanoscience, University of Nebraska, Lincoln, NE 68588, USA. [5] These authors contributed equally: Xuejing Shen, Tao Sun. ✉email: wuzhj@dlut.edu.cn; ltan4@unl.edu

Fast charging is the key feature for portable electronics and electric vehicles which has ignited vigorous research activities. For energy storage platforms that rely on reversible redox reactions, the reduction in charging time from hours to minutes has already become a reality. A typical example can be found in a non-lithium platform, i.e., Al-ion batteries[1]. Over the past five years, it has quickly captured the fame of exceptional rate in both charging and discharging. The adoption of a pure Al as the electrode provides significant merits such as low cost, non-flammability, and high capacity. In addition, a stable Al electrode-electrolyte interface removes the complexity from an interphase layer that is commonly seen in lithium or lithium-ion systems[2,3]. As such, long lasting performance with several tens of thousands of reversible charging and discharging has been demonstrated[1].

Can we further reduce the charging time from minutes to fractions of a second while keeping most of the capacity? We have seen great works from different research groups, where they focused on getting a higher specific capacity[1,4,5], synthesizing a new carbon electrode to promote adsorption[4,6–9], or finding an affordable organic electrolyte[10,11]. Rarely has attention been paid at the intrinsic barrier for charge transfer through the interface between the electrolyte and the electrode. Physics considerations suggest that faster charging requires a larger current injection; but a larger current will result in larger drop in resistance ($iR$) at the interface. From a chemistry standpoint, metal ions in state-of-the-art Al-ion batteries exist as anionic complexes; the rate of reduction for these large negatively charged ions is much slower than the reduction rate of metal salts in water. If the limitation in charge transfer is removed, we can then expect much bigger impacts than mere savings in time. For instance, this will eliminate the clear boundary between a supercapacitor and a battery, making the device both high capacity and high rate; and it will provide a deeper understanding of the electric double layers (EDLs). It has been generally accepted that thin, in the range of a few nanometers, EDLs exist at the interface between electrolyte and a metal electrode. Current research treats EDLs as stable nanostructures[12]. It is currently not clear how EDLs participate in the reduction of negatively charged ions. It is even less known about how to regulate EDLs in order to facilitate a quick reaction at the interface.

In this study, we demonstrate that charge transfer through the interface between Al electrode and the organic electrolyte can be effectively accelerated. As a result, the sites for $Al^{(0)}$ deposition are no longer assisted by surface defects only. We gained multiple technological and scientific advances including the ultrafast charging rate, high capacity, and 500% higher specific capacity under high-rate conditions. Most importantly, acceleration of the charge transfer reaction enabled the discovery of many intermediates inside the EDLs, expanding our understanding of the role that EDLs play in rechargeable batteries. We show that the byproducts formed during charging/discharging can be used to calibrate and challenge conventional understanding in the bulk.

## Results

**Intrinsic barrier in charging**. Al-ion batteries earned their fame by using an organic cation-based electrolyte[1,5], similar to those cases in lithium[13] and lithium-ion batteries[14]. Different from metal salts in water, cations here do not have any metal element; therefore, they don't directly participate in redox reactions. Instead, the metal ions exist as anions or as negatively charged metal complexes. Preparation of the electrolyte is straightforward: mixing imidazolium chloride ($EMI^+Cl^-$) (solid) and anhydrous powder of $AlCl_3$ produces an ionic liquid (eutectic mixture). Three major ions have been reported in this electrolyte, i.e., Al mono-complex ($AlCl_4^-$), Al duo-complex ($Al_2Cl_7^-$), and the organic cation ($EMI^+$)[5,15]. When this electrolyte is placed inside an Al-ion battery, the Al electrode will be biased negatively and carbon electrode positively for charging. As a result, electrons from Al will jump over to the Al duo-complex and reduce it to a mono-complex, depositing fresh $Al^{(0)}$ over the Al electrode. On the carbon side, no new products will form. Rather, the Al mono-complex will adsorb on positively charged carbon surfaces. When batteries are allowed to discharge, Al (anode) will be oxidized but the carbon (cathode) reduced.

We used a three-dimensional (3D) network of graphene as the cathode to promote charge capacity, along with pure Al as the anode. Figure 1a shows the network structure of our graphene, where the carbon-growth on a nickel foam was handled inside a chemical vapor deposition (CVD) chamber[16] (see Supplementary Fig. 1). Later removal of the nickel template requested acid dissolution, solvent rinsing, and drying. We found that the graphene cathode can exhibit smaller redox potentials in cyclic voltammogram (Fig. 1b) only when the drying step is handled using supercritical $CO_2$. Shifted peaks in the voltammogram suggest higher affinity for anions ($AlCl_4^-$) to bind to the surface (Fig. 1a-middle); an open and continuous network would then allow for a reliable desorption (Supplementary Figs. 2 and 3). Seemingly, the graphene cathode acts as an open pocket by holding anions ($AlCl_4^-$) during the charging process. When these anions bind to the graphene (positively biased while charging), three carbon-chloride bonds (Fig. 1a-middle) could form, rendering a robust "holding" of the Al mono-complexes. As strong bonding lowers energy of the system, we hypothesize that further cleavage of these bonds would be energetically costly, making discharge prohibitive under a high rate.

Exposed thin layers from the 3D graphene further improve performance of the Al-ion batteries as shown in Fig. 1c. We first observed a record-high[1,4–9] specific capacity (200 mAh g$^{-1}$) under a current density ($i$) of 20 A g$^{-1}$ ($C$-rate of 100; charging density ($i_c$) same as discharging ($i_{dc}$) or $i_c = i_{dc}$), then the capacity dropped at higher discharge rates ($i \geq 200$ A g$^{-1}$ or rate over 1,000 $C$). Details of these charging/discharging are shown in Fig. 1d. Comparison between Fig. 1e, f further provided reasons for the capacity decline, where reduced capacity retention was partially due to a fast discharging. Namely, when charging rate was kept at a moderate level ($i_c = 100$ A g$^{-1}$) but followed by a fast discharging ($i_{dc} = 100 \sim 600$ A g$^{-1}$), clear loss of capacity in the charging plateau (shortened charging time; Fig. 1e-left) or a widespread quick drop in capacity retention (Fig. 1e-right) was observed. However, when this sequence was reversed, i.e., charging at a really fast rate ($i_c = 400 \sim 1,000$ A g$^{-1}$) but followed by a moderate rate of discharging ($i_{dc} = 100$ A g$^{-1}$), loss of capacity became much less severe (Fig. 1f). Again, these data agree with our earlier statement that the graphene pocket is good at adsorbing anions but does not release them very well. In other words, a densely packed pocket would make the absorption of anions challenging, leading to inferior performances or a reduction in specific capacity (Supplementary Fig. 4). Beside pocket size, we do not foresee any barrier for fast charging at the cathode side, where one-atom-thick carbon layer presents minimal resistance for current injection and Al-mono complex ($AlCl_4^-$) naturally likes a positively charged surface (graphene).

Fast charging at the anode side, however, is not simple. Mainly, Al species inside the organic electrolyte carry negative charges, either as mono-complexed ions ($AlCl_4^-$) or duo-complexed ones ($Al_2Cl_7^-$)[1,5]. The only way to reduce these Al-complexes is to negatively bias the Al anode. This, however, will result in oppositely charged cations ($EMI^+$) adsorbing on the anode first, leaving anions no choice but to adsorb as the second layer. Such two-layered structure will then stack on top of one another multiple times to form the so-called EDLs. Due to the presence of

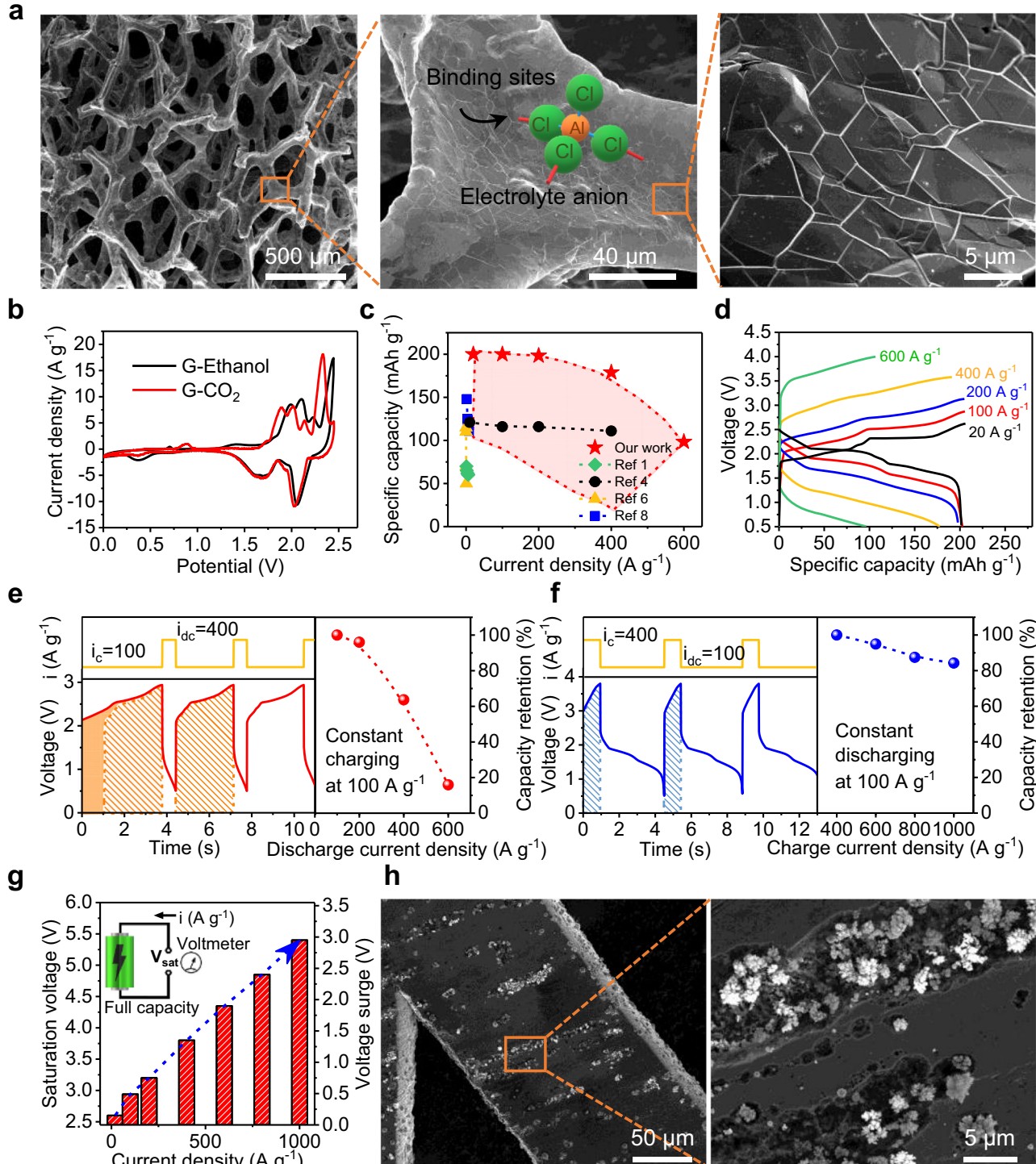

**Fig. 1 Highlights of Al-ion batteries and their performance limits. a** Scanning electron microscopy (SEM) images of a three-dimensional graphene network after supercritical $CO_2$ drying. Large open pores with interconnecting frameworks are clearly visible. **b** Cyclic voltammograms of graphene that was either supercritical $CO_2$ dried (G-$CO_2$) or dried by evaporating ethanol (G-Ethanol) (scan rate of 10 mVs$^{-1}$). **c** Plot of the specific capacity versus current density for our work (entire red block) and state-of-the-art. **d** Galvanostatic charge and discharge curves for devices having record-high specific capacities (200 mAh g$^{-1}$). The graphene cathode has a mass of 0.013 mg and density of 0.16 mg cm$^{-2}$. **e** Fast discharge ($i_c = 100$ A g$^{-1}$, $i_{dc} = 100 \sim 600$ A g$^{-1}$) leads to a quick drop in specific capacity (area of the shadow to assist the view on the amount of charging capacity). **f** Moderate discharge followed after a fast charge ($i_c = 400 \sim 1,000$ A g$^{-1}$, $i_{dc} = 100$ A g$^{-1}$) retains 85% specific capacity even when batteries were charged at 1000 A g$^{-1}$. **g** Charging voltage to maintain a decent specific capacity goes up quickly with the increase of current density. **h** SEM images of spotted Al islands inside surface pits of a pure Al anode after battery cells were fully charged at 400 A g$^{-1}$.

EDLs, electrons from the electrode cannot reach those Al-complexes without tunneling through the $EMI^+$ layer. Scanning tunneling microscopy studies in liquid[12,17] have confirmed such tunneling of electrons through the $EMI^+$ barrier. Therefore, the reduced $Al^{(0)}$ adatoms will need extra amount of energy before being deposited across the same $EMI^+$ layer. Figure 1g confirms the existence of this energy barrier in fast charging. A voltage surge as high as 3.0 V was recorded when a large amount of current was injected through the Al anode. Interestingly, the device used surface defects for $Al^{(0)}$ depositions. Figure 1h shows that flower buds-like Al grew almost exclusively inside the surface pits. This defect-guided growth suggests a reduction in surface energies being adopted to minimize the total consumption in energy. As those buds were spherical in shape, Al plating must have occurred at the same rate in all directions[18]. By increasing the surface energy in Al anode we can therefore push the growth rate of $Al^{(0)}$ further. This can be achieved utilizing liquid metal instead of pure aluminum.

**Increasing surface energy with liquid metal**. Gallium has been reported as a good solvent for aluminum when heated[19]. Galinstan (dubbed as liquid metal or LM), on the other hand, is a eutectic alloy (m.p. −19 °C) of gallium (68.5%), indium (21.5%), and tin (10.0%) (all by weight)[20]. Not only does this alloy inherit the dissolving power from gallium, it lowers the working temperature without the need of heating[21]. At room temperatures, we can dip a piece of Al into a pool of liquid metal. Non-uniform infiltration of liquid metal crossing Al grains will naturally occur after extended period of time (min to hour). Liquid metal will fill the grain boundaries as well as those defect sites (Fig. 2a). Solid Al surface (green stripes) can then transform into a domain that is Al-rich (trace of Ga as pink dots) but still solid-like and another domain that is Ga-rich but liquid-like (pink patch). As the boundaries between both domains are Al-rich (green dots) but highly amorphous, they would act as high-surface-energy sites for subsequent Al plating.

This active anode (Al-LM) is expected to show several advantages. To name a few, the initiation of Al growth will no longer be limited at the defects anymore. Instead, it will grow over the amorphous boundaries everywhere. Next, each nucleation spot can trigger an explosive growth by forming Al dendrites (Fig. 2a). Large surface areas from the dendrites then shall produce even higher surface energies for continued Al deposition. As no solid interphase layer will generate from the electrolyte, these dendrites will maintain an intimate contact with Al-LM. Thus, long-term operation of these devices will not be affected as it does in lithium or lithium-ion batteries[2,3]. In addition to these advantages, Al-LM batteries were found with one more benefit as indicated by the results shown in Fig. 2b-bottom, where high Coulombic efficiencies (~98%) were received immediately after the batteries were installed. In contrast, devices with a pure Al anode gave low efficiencies (~70%) at the beginning (Fig. 2b-top), likely due to an incomplete stripping of the flower bud-like structures. Certainly, if those surface pits were filled with residual buds, continuous charging and discharging would then start to gain high Coulombic efficiencies (~98%). The most exciting benefit with the new anode is that the charging rate can indeed be increased even further (Fig. 2c-left), e.g., $10^4$ C (1000 A $g^{-1}$; charge to full capacity of 88 mAh $g^{-1}$ in 0.35 s). Figure 2c-right shows full cycles of battery operations placed side by side. For the new active anode, not only did the batteries show higher specific capacities (longer time in discharging), their charging plateaus were also much lower (corresponding to smaller voltage; Supplementary Fig. 5). If we now compare specific capacity in both cases with the same charging voltage (Fig. 2d), we see strong

gains in performance, i.e., 5 times more specific capacity (42.2 vs. 7.1 mAh $g^{-1}$). This performance leap confirmed a lowered energy barrier for $Al^{(0)}$ depositing. In other words, a reduction in the interface resistance is highly likely, as evidenced by the electrochemical impedance spectroscopy (EIS). In Fig. 2e, the active anode (red) had 3 times less resistance than the pure Al (blue) (see Supplementary Fig. 6 for the circuit model and data fitting).

We designed two planar devices to record the accelerated growth rate of $Al^{(0)}$. The anode in one device was a piece of Al mesh but the other one having the mesh briefly treated with liquid metal. We placed both devices under an optical microscope and then let them be overcharged under 400 A $g^{-1}$ for extended period of time. As shown in Fig. 2f, early stage of charging already made newly grown Al different, rather small flower buds (top panel) for the first design (pure Al) but extended fractal structures (bottom panel) for the second design (Al-LM) ($t = 1.8$ s, Supplementary Fig. 7). Afterwards, side views suggest small deposits growing into tall deposits, either adopting a dense, brush-like morphology (Al) or as isolated ferns (Al-LM) ($t = 3$ min). Later on ($t = 10$ min), top view revealed another distinction: Al adatoms prefer to nucleate in a flat area but not on existing brushes (pure Al); in contrast, fractal structures on Al-LM kept getting wider and bigger. Once the overcharging was allowed to continue further, those brushes on pure Al eventually became taller or wider ($t = 30$ and 60 min). These consecutive snapshots showed two benefits obtained from the Al-LM anode, one is easier surface nucleation and another is continued reactivity on already-grown deposits. However, as above LM treatment is rather brief (~3 min), we expect more growth sites when treatment time is extended. But how much longer do we need?

**Optimal amount of liquid metal**. To answer this question, we analyzed the surface domains that form as a result of non-uniform infiltration of liquid metal crossing Al grains (Fig. 2a). If we classify the treatment time from short to excessive, we then expect the amount of these reactive sites to increase at first and then decrease. For instance, when the treatment time is short (Fig. 2a-2nd row), a small amount of liquid metal is introduced. Thus, a small portion of the anode surface is modified, with surface pits disappearing first and other areas lightly permeated with gallium. This eventually should produce isolated liquid domains that are surrounded by large patches of solid domains. When the treatment time is extended, more liquid domains and more reactive sites between domains should form (Fig. 2a-3rd row). Clearly, when the treatment time becomes excessive, the liquid domains will connect to form a large and thick patch (Fig. 2a-4th row), with solid domains quickly disappearing and reactive sites sparsely distributed. Either way, dendrites grown on Al-LM must be separated by empty spaces (inactive domains). Therefore, the dendrites are wide but not sharp. This is also the biggest difference we saw between the two cases in Fig. 2f. One interesting feature from these inactive patches, however, is the patch-to-sphere transformation. When reactive sites accept newly deposited Al by forming dendrites, these dendrites will push liquid domains next to them, switching the thin film-like, liquid domain into a sphere or a particle (Fig. 2a). The results in Fig. 3a supported this expectation with additional details. Namely, when the anode was freshly treated by liquid metal in a short time (5 min), we first saw a smooth surface without any pits or cavities (Supplementary Fig. 8). Element mapping revealed that this surface consists of small Ga-rich domains, morphologically similar to surface cavities previously shown in Fig. 1h. Further mapping in the Al-rich domain, on the other hand, uncovered

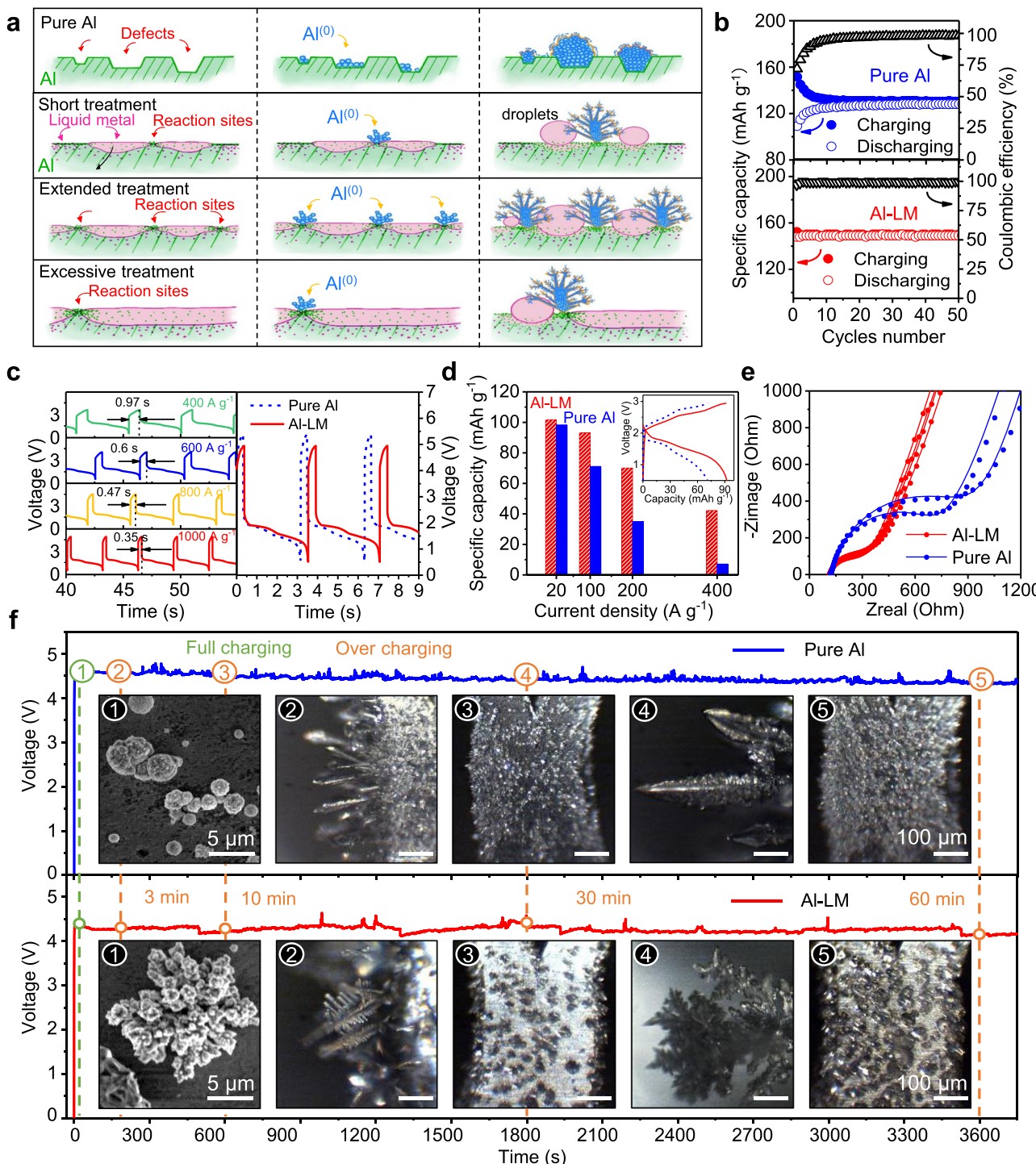

**Fig. 2 Active anode (Al-LM) promotes easy Al plating. a** Schematic illustration of the plating of Al adatoms on pure Al versus that on Al-LM. **b** Active anode behaves differently from a pure Al anode, where no delay in specific capacity and Coulombic efficiency were observed. **c** Al-LM promotes an ultrafast charging with excellent specific capacity ($i_c$ = 400~1000 A g$^{-1}$, $i_{dc}$ = 100 A g$^{-1}$), where a mere 0.35 s can charge the battery to its full capacity. Compared with pure Al anode, the active anode requests a lower charging voltage and exhibits longer time of discharging duration ($i_c$ = 1000 A g$^{-1}$, $i_{dc}$ = 100 A g$^{-1}$). **d** Bar graphs of active anode vs. pure Al anode in producing better specific capacity under high rates. Saturation voltage of Al-LM anode was used as cut-off voltage for both cases. (Inset) Charge and discharge curves at a current density of 100 A g$^{-1}$ (graphene parameters: 0.025 mg; 0.22 mg cm$^{-2}$). **e** Electrochemical impedance spectroscopy (EIS) reveals pure Al anode higher resistance than the active anode. **f** Over-charging of Al-ion batteries with two different anodes (Al vs. Al-LM, $i_c$ = $i_{dc}$ = 400 A g$^{-1}$). ① SEM images of full-charging show early morphologies drastically different (scale: 5 µm); and ②-⑤ are optical microscopy images of front- and side-views of plated Al (scale: 100 µm).

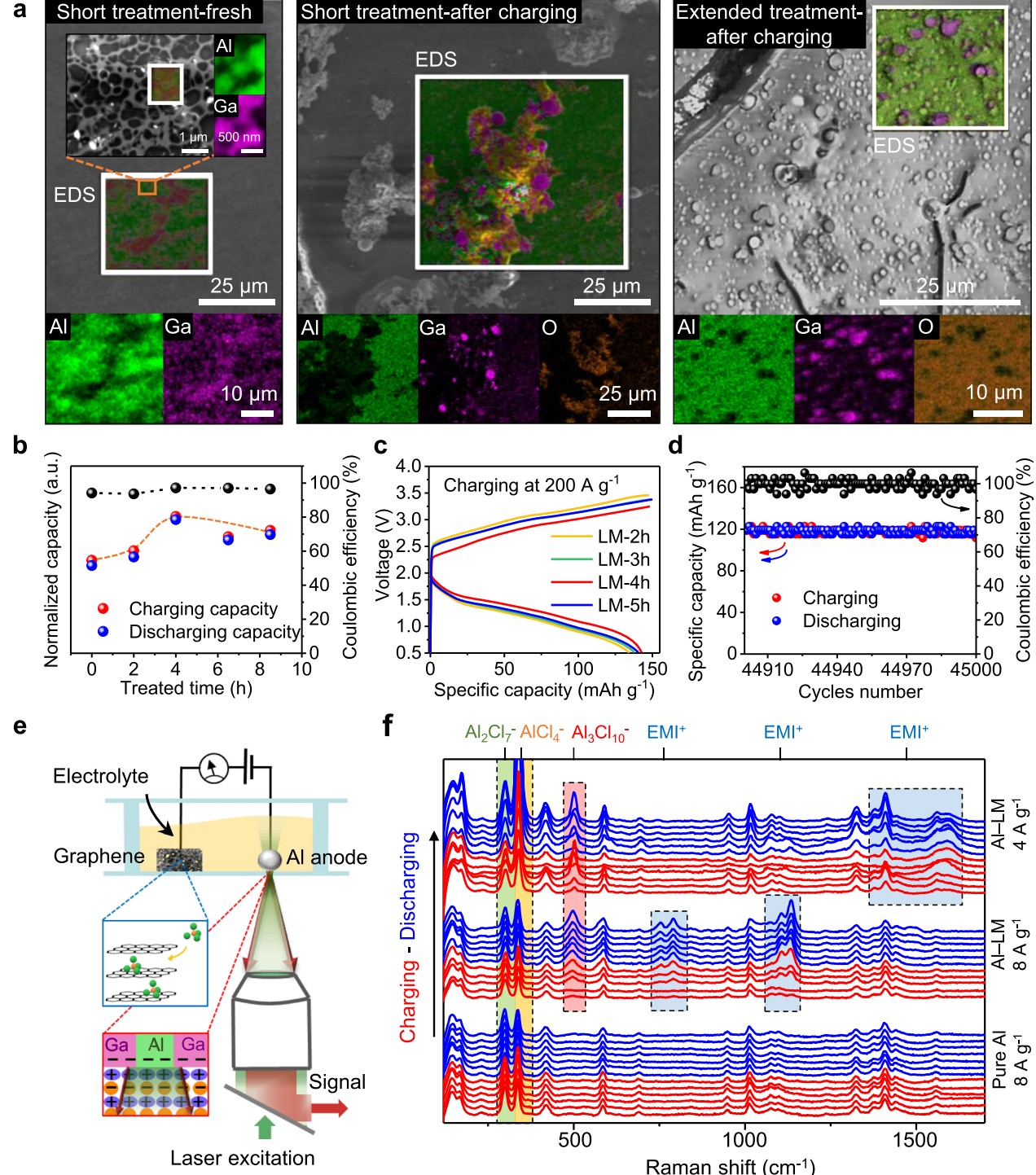

**Fig. 3 Probing the role of the active anode. a** SEM images and elemental mapping (EDS) of gallium distribution on anode. Before charging, liquid metal forms a spread-out network on Al. After charging, part of the liquid metal wraps up as spheres next to those newly grown aluminum sites. **b** The effect of liquid metal treatment time on capacity (charging and discharging current density of 20 A g$^{-1}$ and cut-off voltage of 2.45 V). **c** Galvanostatic charge and discharge curves. Graphene cathode has a mass of 0.026 mg and density of 0.16 mg cm$^{-2}$. Note the optimal time (4 h) has the lowest saturation voltage and maximum capacity ($i_c$ = 200 A g$^{-1}$). **d** Stability test of our Al-ion batteries using active anode over 45,000 cycles (same charging and discharging current density of 40 A g$^{-1}$, cut-off voltage of 2.45 V). **e** Raman setup to study reaction on the active anode. **f** Time series of Raman spectra for one full cycle of charging and discharging at the interface of anode ($i_c$ = $i_{dc}$). Al$_2$Cl$_7^-$, 299 cm$^{-1}$ (green zone); AlCl$_4^-$, 338 cm$^{-1}$ (yellow zone); Al$_3$Cl$_{10}^-$, 500 cm$^{-1}$ (red zone); and EMI$^+$, 753, 790, 1135, 1410, 1590 cm$^{-1}$ (blue zone).

channels of Ga inside polycrystalline Al grains. When this piece of anode was charged in a battery, dendrites were generated, with Ga-rich (purple) spherical particles lying next to the roots. While we did not detect signals from oxides on a freshly treated anode,

dendrites from a charged anode were different: a brief exposure in air made them oxide rich (seconds before sealing the SEM chamber), while surrounding flat domains were not much affected by this exposure. Once the anode treatment was extended to

hours, modified surface after charging then exposed a large number of Ga-rich particles, largely supporting earlier expectation on patch-to-sphere transformation. Figure 3b depicts the dependence of specific capacity on treatment time, where new anode indeed had better performance in high rate operations and an optimal value was obtained after a treatment of 4 h. Figure 3c displays multiple performances laid on top of each other, showing the active anode of 4-h by Galinstan having the lowest charging plateau and the longest discharging time ($i_c = 200$ A g$^{-1}$). Intriguingly, aforementioned droplets or particles shown in Fig. 2a had no interference in the repetitive charging/discharging. Rather stable operations were recorded when the device was cycled for 45,000 times (Fig. 3d).

**Reaction intermediates next to active anode.** We used Raman spectroscopy to track the events at the anode surface. High intensity Raman signals are expected due to the surface plasmon effect in Al electrode[22]. Rich production of transient intermediates during charging-discharging also contributes to relatively intense and interpretable Raman signals. In Fig. 3e, a battery with a planar configuration was sealed and placed over a glass coverslip, where the reaction on anode was monitored with a laser excitation ($\lambda = 532$ nm) through the coverslip. By comparing the intensities of Raman signals measured in the bulk electrolyte and measured when aluminum anode was excited, we estimate the Enhancement Factor to be EF = 11.5. The intensity of Raman signals strongly depends on the intensity of local electric field because of the surface plasmons in aluminum electrode. Due to evanescent character, the intensity of electric field falls off exponentially with distance away from the anode, penetrating a very short distance (~nm) into the surrounding medium[23]. This allowed us to selectively probe events happening primarily next to the active anode.

Figure 3f shows the Raman spectra throughout the charging-discharging cycle. Three panels illustrate three scenarios. Spectra shown in the bottom panel suggest that when the anode is made out of pure Al all the peaks corresponding to aluminum complexes and EMI species remain stable except for those at 299 and 338 cm$^{-1}$ which respectively belong to Al$_2$Cl$_7^-$ and AlCl$_4^-$. The intensities of both peaks change throughout the cycle, with the ratio ([AlCl$_4^-$]/[Al$_2$Cl$_7^-$]) depicted in Supplementary Fig. 9. This trend matches well the existing general notion[1] of the reaction taking place described using the following equation:

$$4\text{Al}_2\text{Cl}_7^- + 3\text{e} \leftrightarrow 7\text{AlCl}_4^- + \text{Al} \qquad (1)$$

Surprisingly, we found that the reaction species adjacent to the Al-LM (Fig. 3f-top, middle) are different from those next to pure Al. With Al-LM not only do we see transient intermediates for EMI$^+$ but also Raman signatures corresponding to a triple-complex of aluminum (Al$_3$Cl$_{10}^-$). It is worthwhile to note that the rate of the peak disappearance does not exactly follow the rate of discharging. Rather, it takes much longer time for these peaks to fully disappear. As these peaks are captured over the surface of active Al-LM electrode, but not pure Al electrode, we propose that Al-LM electrode differs from Al as much as to allow for the intermediate triple-complex to easily form. Further analysis of the reaction mechanism will help us answer the following questions: How would a new anode accelerate the Al-deposition? And how did this acceleration disrupt the conventional structure of EDLs?

**Preferential nucleation on active anode.** Among the three elements in Galinstan, gallium is the major component and also the only element that plays a pivotal role in lowering the redox potential in Al electroplating (see Supplementary Figs. 10 and 12).

While the formation of surface domains back in Fig. 2a seems reasonable to account for this potential lowering, very little is known about why the boundaries inside the active anode are more reactive. With partial coverage of Al surface by Ga we expect a strong effect of Ga presence on both adsorption and diffusion of the Al adatoms. We investigated the preferential nucleation location on such a composite surface, taking into consideration the adsorption energy differences in the first approximation. We calculated the adsorption energy of Al adatoms on Al(111) and compared it with the respective value on Ga monolayer covering Al(111). The results shown in Fig. 4a indicate that the adsorption on pure Al surface is much more favorable (away from the Al/Ga interface or boundary).

However, we expect the Al/Ga interface will have several nucleation spots. Particularly, Ga is expected to form islands either on the planar Al surface or fill Al surface imperfections such as cracks and scratches. We used DFT calculations to analyze the two configurations: (1) a large Ga patch on Al(111); and (2) a small Ga island covers a small cavity in the Al surface (three high symmetry surfaces (111), (100) and (110)). When a Ga island covers a small cavity (~3–4 interatomic distances) on Al surface, our calculations (details see Supplementary Fig. 13) show that Al adsorption energy near the interface of such a planar surface could be lower than that on pure Al(111). The adsorption energies, however, are more complicated with a Ga patch. We analyze with alternating strips of Al and Ga monolayer. Figure 4a shows the adsorption energies calculated for hcp (H), fcc (F), and the bridge position (B) between the first two sites. The first conclusion we can make is that, the adsorption energy is not a monotonic function of the distance from the boundary between Al/Ga. There is a sharp increase in adsorption energy right next to the boundary. Far from the interface there is a much larger adsorption energy on the Ga monolayer. Thus, energetically favorable adsorption near the Al/Ga boundary is highly possible and this will lead to preferential sites for nucleation. Then, we compare the interatomic distances (bond lengths) for adsorbed Al in terms of Al-Al and Al-Ga pairs across the Al/Ga boundary. Results shown in Fig. 4b-right indicate that Al in H4 position is indeed more favorable, due to a stronger Al-Ga bonding (Al-Ga bond length decreases to ~2.6 Å compared to 2.625 Å at monolayer coverage). Meanwhile, differential charge density exhibits a strong localization of electrons around the Al-Ga pairs, where the formation of bonds with adatoms is accompanied by a noticeable disruption in Ga-Ga surface bonding (it gets almost zero in differential charge density). In comparison, the H3 position has a much higher absorption energy, with bonding details shown in Fig. 4b-left. Energetically unfavorable bonding between Al adatom and the H3 position is evidenced by longer interatomic distances (d$_{\text{Al-Ga}}$ ~ 2.63 and d$_{\text{Al-Al}}$ ~ 2.67 Å, all larger than Al adatom on pristine Al(111)). Bonding of Al adatom in H3 position is more delocalized, but there is no significant change in surface differential charge density. In other words, adatom at the H3 position will not redistribute to form new bonds with neighboring Al and Ga atoms.

Next we explain the low barrier at the bridge position between the fcc and hcp sites. Mainly, not only can the Al adsorbing on Ga strips (B44 position in Fig. 4a, c) form bonds with two nearest bridge atoms (d$_{\text{Al-Ga}}$ ~ 2.58 Å), it can also bond with two other Ga atoms along the orthogonal direction (d$_{\text{Al-Ga}}$ ~ 2.76 and 2.95 Å). As the bonds along this orthogonal direction are weaker, these Ga atoms could elevate slightly from the surface and move closer to adsorbing Al with distances shortened to Al-Al distance in the bulk (2.87 Å). That is to say, having four bonds is more energetically beneficial than maintaining a 3-fold symmetric adsorption site with 3 nearest atoms.

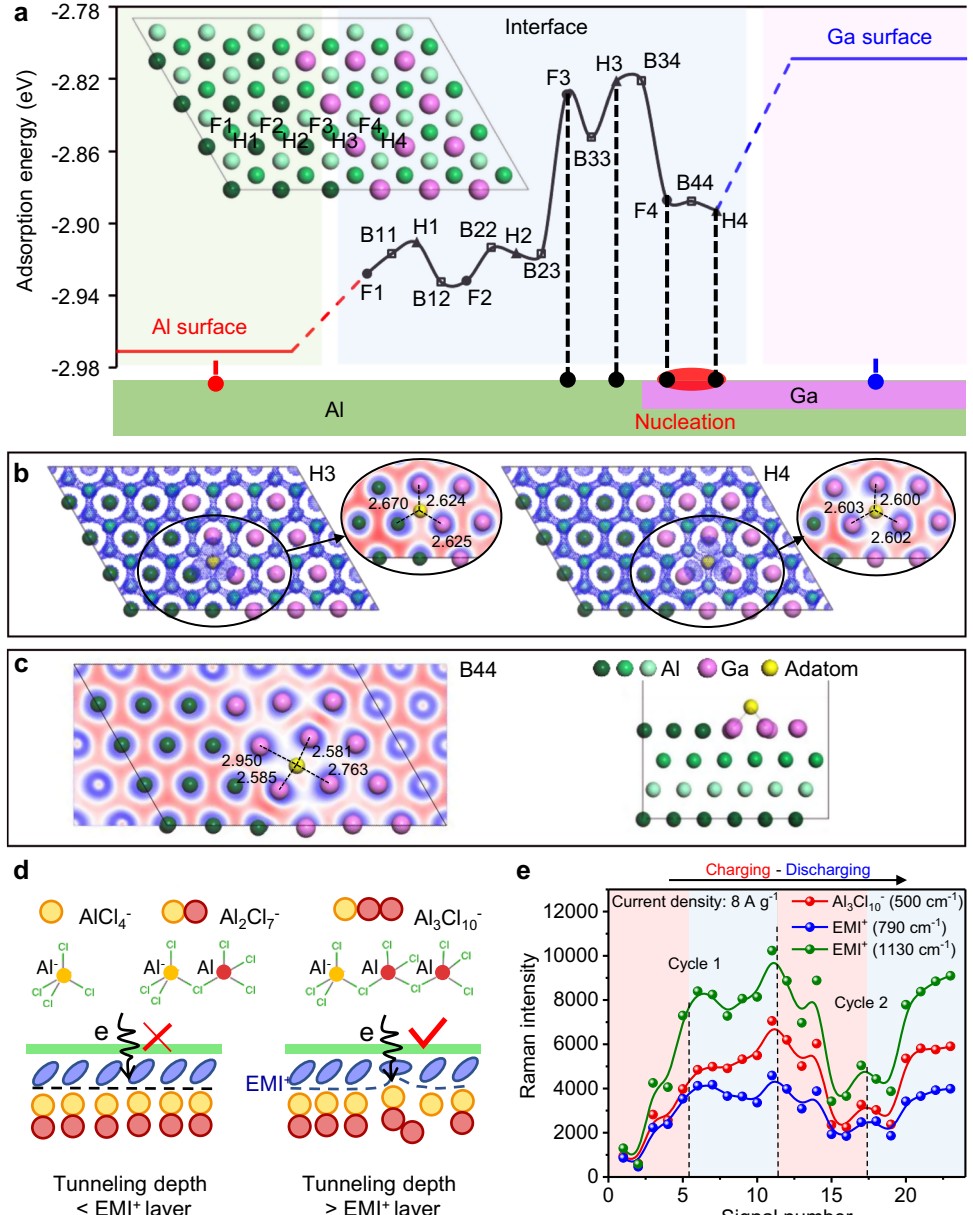

**Fig. 4 Density-functional theory (DFT) calculations reveal the nucleation sites of Al adatom and dynamic nature of the electric double layers (EDLs). a** Adsorption energy of Al on different hcp (H), fcc (F), and bridge (B) positions of Al/Ga interface compared with on pure Al and Ga surfaces. The energetically favorable adsorption near the Al/Ga boundary creates a potential nucleation site. **b** Differential charge density of H3 and H4 adsorption sites. The H4 site exhibits somewhat stronger localization of electrons at the Al-Ga bond, accompanied by the formation of bonds with adatom, making it the most favorable adsorption site. **c** The B44 site shows the disappearance of barrier at bridge position between fcc and hcp sites due to the lowering of local symmetry near the interface. **d** Schematic illustration of the dynamic transition in EDLs. Reaction intermediate ($Al_3Cl_{10}^-$) triggers reconfiguration for $EMI^+$. **e** the intensity variation with time for $Al_3Cl_{10}^-$ and $EMI^+$ indicates a coordinated change for both ion species during charging (within red shade) and discharging (within blue shade).

The above analysis was performed on Al(111) surface where adatoms are 3-fold coordinated and diffusion barrier for Al self-diffusion is trivial. Similar conclusions can be made for Al(100) and (110) surfaces containing Ga islands (see Supplementary Fig. 14). The coordination of Al atom on the surface changes in the presence of Ga. For example, Al acquires two extra neighbors when attaches to the Ga island which may serve as a nucleation site both at (100) and (110). Especially drastic observation is received for Al(110) case. The lowest energy position is at the Ga island site because Al binds not only to Ga but also subsurface Al neighbors. This increases the overall adsorption energy. Thus, the ability of Ga atoms to promote an additional bonding with Al

adatoms make it a perfect "surfactant" to augment the growth kinetics.

The above calculations assume that there are no strong interactions with molecules of the ionic liquid. Such interactions could come during electroplating. We investigated an effect of ionic liquid on a bridge-hopping diffusion process for (100) surface (see Supplementary Fig. 15). Although the adatom bonding with the ionic liquid molecules changes, the strength of interaction with ionic liquid is order of magnitude smaller than the interaction of adatom with the substrate. As a consequence, earlier approximation to explain the contribution from the Ga coverage on Al deposition is adequate.

The energy landscape of the Al diffusion support the nucleation and growth process described above and illustrated in Fig. 2a. Ga strongly modifies the surface morphology making native defect sites inaccessible for Al growth (preventing low Coulombic efficiency). Al diffuses away from Ga-covered surface towards the free Al surface and nucleates at the Al-Ga disordered interface of Ga-free surface. Thus, the directed diffusion increases Coulombic efficiency and prevents the passivation of the electrode due to the multilayer coverage (observed, for example, in underpotential deposition conditions[24]).

**Possible new reaction route.** Electric double layers (EDLs) next to the active anode (Al-LM) are likely to adopt a lamellar structure like any other electrochemical systems with an organic electrolyte. Current research in surface science treats EDLs as stable nanostructures. This includes revealing them as lamellar stacks[12], interpreting the layered formation with the concept of overcompensation in charge[25], and capturing nonuniformity over topography defects[26]. Reported studies from the electrochemistry community mainly focused on bulk reactions. It is generally assumed that the reaction mechanism appropriate for the bulk should apply to the EDLs too. Rate acceleration, we achieved herein, offered us an opportunity to look into the reaction along the electrolyte-electrode interface.

New peaks in Fig. 3f represent the reaction byproducts at the nanometer vicinity of the active anode (Al-LM). Not all of them, however, are accounted for in the conventional charging mechanism (Eq. 1), i.e., $4Al_2Cl_7^- + 3e \rightarrow 7AlCl_4^- + Al^{(0)}$. To account for all the observed byproducts, instead of one-step conventional reaction, where electrons from the anode directly reduce 4 parts of Al duo-complex ($Al_2Cl_7^-$) to $Al^{(0)}$, we propose the existence of two extra steps. The first step starts from a subtle change in EDLs. Here, reorganizing two neighboring Al duo-complexes can produce a triple-complex and a mono-complex (Eq. 2a). Since the triple-complex is larger than duo-complex, it may disrupt the uniformity of the organic cationic layer in EDLs (Fig. 4d). In other words, appearance of a large Al complex will prompt the rearrangement of EMI cations. When EMI cations are forced into a different configuration they will stay closer to the electrode (Eq. 2b) which, in turn, will facilitate tunneling of electrons to the large triple-complex assisting in deposition of Al$^{(0)}$ (Eq. 2c – with triple-complex the only reactant or 2d – with duo-complex as additional reactant):

$$2Al_2Cl_7^- \leftrightarrow Al_3Cl_{10}^- + AlCl_4^- \qquad (2a)$$

$$EMI^+ (standing\ up) \rightarrow EMI^+ (lying\ down) \qquad (2b)$$

$$2Al_3Cl_{10}^- + 3e \rightarrow 5AlCl_4^- + Al^{(0)} \qquad (2c)$$

$$Al_3Cl_{10}^- + 2Al_2Cl_7^- + 3e \leftrightarrow Al^{(0)} + 6AlCl_4^- \qquad (2d)$$

This new reaction route above is supported by the signature of the new peaks in Fig. 3f, in which dihedral angle torsion (753 and 790 cm$^{-1}$) and C–C/C–N bond stretches (1135, 1410 and 1590 cm$^{-1}$) resemble peaks observed for the compressed organic cations (EMI$^+$)[27,28]. The aluminum triple-complex ($Al_3Cl_{10}^-$), on the other hand, generates the peak at ~500 cm$^{-1}$. If we single out the new peaks from the current density of 8 A g$^{-1}$ (Fig. 3f) by plotting their intensities vs. the charging/discharging sequence as in Fig. 4e, correlated intensity changes of these intermediates are clearly evident (see Supplementary Fig. 16 for coupling of intermediates under the current density of 4 A g$^{-1}$). This again supports the proposed reaction steps from Eqs. 2a to 2c or 2d. It's worthwhile to point out that the Raman intensity fluctuations of the Al triple-complex are observed for different charging cycles.

Such variation of the sensitivity in Raman detection of species is attributed to the formation of dendrites over the active anode surfaces. High degree of dendrites' structural diversity crossing multiple length scales (from nanometer to micrometer) could largely contribute to variability of enhancement factors over cycles of battery operation (see detailed discussions in Supplementary Fig. 17).

**Discussion**

Apparently, the capture of Al triple-complex over the interface of electrolyte and the anode has challenged the conventional understanding in Al-ion batteries. One would question how frequently this new intermediate will form in current densities beyond 4 or 8 A g$^{-1}$ and what role it plays in discharging. We have further created a more active Al-LM anode by soaking a piece of Al wire in liquid metal beyond the treatment time used above (Al-LM$_{HIGH}$: 6 h; Al-LM$_{LOW}$: 4 h) and performed Raman measurements over a wide range of current densities (from 0.25 to 160 A g$^{-1}$, see Supplementary Fig. 18). Extensively treated Al-LM anode did offer a perspective on all participating Al-complexes including single ($AlCl_4^-$), double ($Al_2Cl_7^-$), and triple ($Al_3Cl_{10}^-$) complexes. First, Al single-complex dominates under a small current density, while Al double-complex dominates under a high current density. Second, higher degree of variability in Raman intensities is observed at the intermediate current densities. This observation further corroborates the data shown in Fig. 3f but also points to a complex dependence of Raman intensities on current density and the nature of the interface (Al vs Al-LM$_{LOW}$ vs Al-LM$_{HIGH}$). Additionally, we have observed that Al triple-complex is always formed for the Al-LM$_{HIGH}$ electrode. Triple-complex does no longer disappear completely but varies in intensity, for all the current densities. We therefore postulate a reasonable explanation for Al-triple complex to account for all these new observations as:

$$Al^{(0)} + 2AlCl_4^- + 2Cl^- - 3e \leftrightarrow Al_3Cl_{10}^- \qquad (3a)$$

$$2AlCl_4^- \leftrightarrow Cl^- + Al_2Cl_7^- (EMI^+ assisted) \qquad (3b)$$

The combined reaction involving the triple-complex is as following

$$Al^{(0)} + 6AlCl_4^- - 3e \leftrightarrow Al_3Cl_{10}^- + 2Al_2Cl_7^- \qquad (3c)$$

Note Eq. 3c is the same as Eq. 2d when the latter runs in opposite direction (i.e., discharging). Equation 3a and 3b provide a simpler view on discharging reaction than the conventional one (Eq. 1: $Al^{(0)} + 7AlCl_4^- - 3e \leftrightarrow 4Al_2Cl_7^-$) for several reasons: (a) a clear connection among all complexes (single-, double-, and triple-) is built; (b) the role of organic electrolyte (EMI$^+$AlCl$_4^-$) in the reaction is further clarified, i.e., it provides Cl$^-$ and frees EMI$^+$ from the cation-anion pair; and (c) it shows clearly where the oxidized Al (Al$^{3+}$) is going, i.e., it inserts between two Al single-complexes and grabs two free Cl$^-$ from the organic electrolyte. A schematic sketch to illustrate these reactions is provided in Supplementary Fig. 19. From descriptions above, we can hypothesize that the Al-triple species could be both short- and long-lived depending on how active the electrode is and what stage the electrode is at (charging vs. discharging).

It is important to note that, for the new reaction in Eq. 2a to take place, there are two prerequisites. First, the spatial gap between the two duo-complexes ($Al_2Cl_7^-$ or $AlCl_3 \cdot AlCl_4^-$) needs to be small, i.e., less than the van der Waals distance of 5 Å for organic molecules[29]. Such that, a small shift for AlCl$_3$ from one of the duo-complex to its neighbor can transform the latter anion to a triple-complex ($AlCl_3 \cdot AlCl_3 \cdot AlCl_4^-$). This tight gap further suggests the anionic portion of the EDLs being internally

organized more like polymer patches. Inside an individual patch, the Al duo-complex can be regarded as the repeating unit in a conjugated polymer, with much-needed flexibility to reorganize into larger complexes for fast charging. Secondly, fast charging may not be the only route to produce those Al triple-complexes. In particular, the new anode (Al-LM) while providing much needed high current densities also results in more frequent formation of the triple-complex. Specific details of the new anode's contribution await further explorations. This includes a careful tuning of the surface composition on Al-LM and evaluate its influence to Raman signals.

Overall, we have made substantial progress first by demonstrating ultra-fast charging Al-ion battery and then by expanding our understanding of the role active anode supporting the EDLs plays in charging/discharging. Performance highlights of our device include: (1) highest reported capacity of 200 mAh g$^{-1}$, where conventional Al-ion batteries[1,4–10] have a value no more than 120 mAh g$^{-1}$. This improvement is achieved with an open network of graphene that has a low redox potential; (2) fastest charging rate of $10^4$ C (1000 A g$^{-1}$; duration of 0.35 sec to reach the full capacity) among all metal and metal-ion batteries[30,31]. It was made possible by keeping the discharge at a moderate level (100 A g$^{-1}$; rate of 1000 C), where adequate ion supplies were ensured by desorption of electrolyte from the graphene cathode; and (3) 500% more specific capacity under high rate operations. Exceptional high rate in charging would cause a large voltage surge at the electrolyte–anode interface and results in low specific capacity; active anode alleviates this surge, with an easier formation of Al adatoms along the Ga/Al boundaries. We expect devices with Al-LM as the anode eliminates the gap between a supercapacitor and a battery. Therefore, devices with other novel cathodes[4,6–9] can all be used to quickly store energy when powerline dropping is expected in a fixed schedule or unexpected with a short notice. This includes energy backup for electric buses that are running between stations, restart a suddenly stopped elevator, or even to minimize power-off-induced loss in manufacturing or production lines.

One area in our future plan is to investigate the Al deposition in the presence of organic electrolyte. Special attention will be given to the proper analysis of electrostatic interactions with non-uniform surfaces, as these features usually show strong non-local character at the interface of ionic liquids and solids[32–35]. To push the high-rate operation further, it is imperative to evaluate the insertion of metal cations ($Al^{3+}$) directly in EDLs, as another boost in charging rate. Not only will it replace those inert organic cations ($EMI^+$) by skipping the energy request on electron tunneling, it will also add a 3-electron process to the total reductions[36,37].

## Methods

**Chemicals and materials**. They were purchased from the following vendors unless otherwise specified: hydrochloric acid (HCl, 37 wt%), toluene ($C_7H_8$, >99.5%), and 1-ethyl-3-methyl-imidazolium chloride-aluminum chloride (AlCl$_3$ : EMI-Cl = 1.5) from Sigma-Aldrich; anhydrous ethanol ($CH_3CH_2OH$, 94–96%), anisole ($C_7H_8O$, 99%), and aluminum wire (1.0 mm in diameter, 99.999%) from Alfa Aesar; acetone ($C_3H_6O$, 99.5%) from VWR BDH Chemicals; poly(methyl methacrylate) (PMMA 950 A11) from MicroChem; epoxy resin (Gorilla™) from Walmart; colloidal silver (60% silver content) from Electron Microscopy Sciences; Galinstan™ (alloy of gallium, indium, and tin) from Consolidated Chemical & Solvents LLC; nickel foam (1.6 mm in thickness, 0.1 mm in diameter, purity > 99.9%) from Alantum Advanced Technology Materials (Dalian) Co., Ltd.; aluminum mesh (55 μm in thickness) from MTI Corporation; silver plated wire (26 gauge; Beadalon™) from Michaels; and copper wire (22 gauge) from Arcor Electronics. Above materials and chemicals were all used as received without further purifications.

**Preparation of 3D graphene cathode**. Large-area, three-dimensional (3D) graphene was grown by chemical vapor deposition (CVD) using a gas mixture of

hydrogen and methane and by placing a nickel foam inside a home-built quartz tube furnace. At first, the Ni foam was cleaved into a narrow strip ($17 \times 40$ mm$^2$) and then thoroughly rinsed with the following solvents: toluene, acetone, copious de-ionized (DI) water, and anhydrous ethanol. After drying, the Ni foam was loaded into the quartz tube and pumped to a base pressure of 10 mTorr. Subsequently, a constant flow of H$_2$ (7.4 standard cubic centimeters per minute or s.c.c. m) was introduced into the chamber, and the tube was heated to 1000 °C and maintained for 20 min, followed by another elevated heating to 1100 °C and a constant flow of methane (20.2 s.c.c.m) to trigger the growth of the 3D graphene over Ni foam. The entire growth process lasted 60 min, after which the furnace was cooled down to room temperature over an hour (details see Supplementary Fig. 1). Resulting 3D graphene / Ni foam was then dip-coated with a thin layer of PMMA (4 wt% PMMA solution in anisole) and baked at 95 °C for 4 h. The PMMA/3D graphene/Ni foam was then cut into small pieces with desired dimensions. Afterwards, these pieces were placed in a HCl bath (3.0 M, 70 °C) for 4 h to completely dissolve the Ni layer and later soaked in DI water (5 times) to remove the inorganic residue.

**Supercritical CO$_2$ drying**. Above PMMA/3D graphene sample was soaked in acetone (6 times) at 50 °C for 1 h, then being placed in anhydrous ethanol and later transferred to a supercritical CO$_2$ dryer (Samdri-780A, USA) where its small chamber was preloaded with 20 mL anhydrous ethanol. Liquid CO$_2$ was pumped into the chamber to keep the pressure at 850 psi. The temperature was kept at 10 °C and purged for 3–5 min. A heater was then used to raise the temperature and pressure in the chamber respectively to 31 °C and 1250 psi (for 4 min). Finally, the pressure in the small chamber was released, and the 3D graphene was recovered.

**Preparation of the active anode (Al-LM)**. Al wire/mesh was cut into desired dimensions. A copper wire was then used as current collector, with the Al part washed with toluene, acetone, DI water, and anhydrous ethanol before being transferred into the glove box. Al wire/mesh was immersed in Galinstan (Al wire for 2–9 h, but Al mesh no more than 5 min). After removal, these alloys were gently wiped off the excess liquid metal on the surface and kept for further studies.

**Battery configuration**. All cells were assembled in the argon-atmosphere glove box (Vacuum Atmospheres) and packed in screw-thread vials (4 mL). These cells use 3D graphene as the cathode (areal loading ranged from 0.16 to 0.22 mg cm$^{-2}$, see additional data in Supplementary Fig. 4), Al or Al-LM as anode (20 mm in length for wire and 5 mm × 10 mm for mesh), and 1-ethyl-3-methylimidazolium chloride/aluminum chloride (1.2 mL) as electrolyte. For the cathode, we use a silver-plated wire as the current collector, with colloidal silver as the adhesive and epoxy the fixing layer.

**Electrochemical measurements**. All measurements were performed outside the glovebox after the battery being sealed with an air-tight cap. Multi-cycled, galvanostatic charge/discharge were carried out on a battery testing system (Neware, BTS-4008, 5 V 50 mA; minimum data interval: 0.1 s). For extremely fast charge/discharge tests, since the number of points collected has a great impact on quantified device performances (for details see Supplementary Fig. 20), these tests were performed on an electrochemical analyzer (CH Instruments, CHI6062E; minimum data interval: 0.1 ms). Specific capacity data reported here are all based on the mass of graphene only. Cyclic voltammetry (CV) was also operated on CHI6062E with scanning ranging from 0 to +2.45 V (scan rate of 10 mVs$^{-1}$). We use 3D graphene as the working electrode and Al as the counter and reference electrode (shown in Fig. 1b). Electrochemical impedance spectroscopy (EIS) measurements were performed using a Gamry Interface 1000E potentiostat in two-electrode mode. The cell was designed to have three electrodes where two of them are anodes (one Al-LM anode and another pure Al) and third one is the 3D graphene cathode. We either use Al-LM/graphene pair or the Al/graphene pair, to minimize the influence of graphene cathode. Frequency range is set from 0.1 to 100 kHz and the AC voltage at 5 mV. In the same cell, we alternate the use of both anodes to ensure minimal aging effects. Each pair of electrodes was charged/discharged in the same electrolyte 10 times before the EIS measurement (Fig. 2e in the main text, see Supplementary Fig. 6 for model and fitting).

**Overcharging**. We placed an optical microscope (MEIJI ML8530) in the glove box and used a digital camera (Tucsen H Series) to record the images via a laptop computer. The battery cell was assembled horizontally on a glass slide, with glass spacers to seal the electrolyte and both electrodes. The cell was placed under the lens of microscope, followed by cycling 50 times between +2.45 and +0.5 V prior to an overcharging test. All overcharging tests were conducted under a constant current density of 400 A g$^{-1}$.

**Structure and morphology characterizations**. The structure of 3D graphene was characterized by scanning electron microscopy (SEM; FEI Nova NanoSEM™ 450), Raman spectroscopy (Renishaw inVia Raman microscope, excited by a 633 nm laser with a laser spot size of 0.3 μm) and X-ray diffractometer (XRD, SmartLab

Diffractometer, Rigaku, Texas, with a CuK wave). For X-ray diffraction (XRD) analyses, the battery cells were repetitively charged and discharged at a current density of 20 A g$^{-1}$. After 1000 cycles, the 3D graphene was removed from the cell. To avoid reaction with the moisture or oxygen from the air, the cathode was placed on a glass slide and then wrapped by a Scotch tape before XRD measurements out of the glove box. Elemental mapping of Al or Al alloy anodes was conducted via an energy-dispersive spectrometer (EDS) attached to FEI Nova NanoSEM$^{TM}$ 450. Fully charged Al or Al alloy anodes were washed with anhydrous toluene to remove any residual electrolyte. Then they were adhered over a carbon conductive tape and sealed in a plastic box before any characterizations.

**Raman measurements**. Raman measurements were performed using a Raman spectrometer configured in transmission mode on the Olympus IX71 inverted optical microscope. An oil immersion Olympus objective lens with 100X magnification and 1.4 NA (UPLSAPO) was utilized for focusing the laser on the surface of anode before collecting the Raman signal. Glass coverslips windows were created in a home-made sealed chamber. The chambers were placed on the stage equipped with an x-y positioning piezoelectric controller. Two experimental setups were utilized: (1) Ntegra-Spectra (NT–MDT, Moscow, Russia) and (2) Raman-HR-TEC (StellarNet, Inc., Tampa, USA). (1) Ntegra-Spectra was utilized for initial detection of reaction species on the anode. Ntegra-Spectra detects the Raman signal using an Andor–CCD camera cooled to −60 °C and optically coupled with both the Raman spectrometer and inverted microscope. A diode laser with $\lambda = 532$ nm and a nominal power of 100 mW was used for excitation (LaserExportCo, Ltd, Moscow, Russia). (2) Raman-HR-TEC (StellarNet, Inc., Tampa, FL, USA) was utilized for automated fast collection of spectra at various charging densities. The spectrometer is coupled to both the inverted microscope and the laser ($\lambda = 647$ nm and a nominal power of 150 mW) using the Raman Probe—the fiber optics cable (StellarNet, Inc.) which integrates both excitation and collection cables. Home-built LabView (National Instruments, Austin, TX, USA) interface in "time series" mode was utilized allowing for collection of spectra without delays at various current densities especially suitable for signal collection at fast rates. Each spectrum was collected for a total of 5 s acquisition time and background corrected for both instruments.

**Computational methods**. Self-consistent electronic structure calculations were performed for the Al/Ga system. The calculations are carried out using the Density-functional theory (DFT) method[38,39] as implemented in the Vienna ab initio simulation package VASP[40]. Projector augmented wave (PAW) pseudopotentials were used[41]. The generalized gradient approximation (GGA) of Perdew-Burke-Ernzerhof (PBE) form[42] is used for the exchange-correlation function. The Al diffusion barrier on Al (111) in a 4-layer slab geometry was selected. The supercell approach was used, with an array of $4 \times 6$ primitive cells arranged in the x-y plane when considering "strip"-like Ga layers on top of Al (111), and $5 \times 5$ array for the case of Ga "island" (Supplementary Fig. 13). We set the plane-wave-cut-off energy to 350 eV and choose the convergence criteria for energy of $10^{-6}$ eV. Calculations were performed with relaxation of atomic positions of all atoms in the unit cells using Hellmann-Feynman scheme till forces were less than 0.003 eV/Å.

## Data availability

All data supporting this study and its findings within the article and its Supplementary Information are available from the corresponding authors upon reasonable request.

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

## Acknowledgements

L.T. gratefully acknowledges the financial support from the University of Nebraska System Science Research Program, Nebraska Center for Energy Science Research, and National Science Foundation (IIA1338988). The research was performed in part in the Nebraska Nanoscale Facility: National Nanotechnology Coordinated Infrastructure and the Nebraska Center for Materials and Nanoscience, which are supported by the National Science Foundation under Award ECCS: 2025298, and the Nebraska Research Initiative. Z.W. gratefully acknowledges the support from National Key R&D Program of China (Grant No.2018YFA0702800), National Natural Science Foundation of China (Grant No. 11902063, U1837205 and 11772075), and Fundamental Research Funds for the Central Universities (Grant No. DUT16ZD214, DUT19ZD101 and DUT19JC32). X.S. would like to thank the lab director (Prof. Ming Li, School of Chemistry and Chemical Engineering, Southwest University, Chongqing, China) for allowing her to perform e-chem tests and You (Joe) Zhou from Nebraska Center for Virology for confocal laser scanning microscopy on dendrites growth/dissolution during the manuscript revisions at the pandemic era.

## Author contributions

L.T. and Z.W. conceptualized the work. X.S. and T.S. made equal contributions in experimental work. M.S. and A.K. assisted in impedance measurement and Raman microscopy, respectively. L.Y., R.S. and W.M. helped with molecular dynamics modeling on anode interfaces. L.T. wrote the first draft of the manuscript. All authors have given approval to the final version of the manuscript. X.S. created all illustrations in the manuscript and Supplementary Information.

## Competing interests

The authors declare no competing interests.
