## [Peer Review File · Nature Communications]

Reviewer #1 (Remarks to the Author):

The manuscript by Shen et al. presents a study on the understanding of ultra-fast charging behavior in aluminum-ion batteries which employ highly active anode (Ga-Al) and 3D grapheme cathode. The ultra-fast charging performance of the Al-ion batteries is impressive and efforts are devoted to elucidate the roles of Ga-Al boundary and electric double layer at the electrode-electrolyte interface, which would be of interest to the large readership of battery and electrochemistry communities. However, there are a number of deficient in the mechanistic aspects of study that makes the work not sound for publication in Nature Comm. Comments are listed as follows.

1. The authors presented two points of views in the introduction section: Pure Al anode is resistive to electron transfer and hinders the production of Al adatoms, and preferential nucleation at defects cast additional barrier for ultra-fast operation, which serve as the origin of problems for the follow-up study of the manuscript. However these views are rather unusual because for Al electrodeposition on pure Al no considerable nucleation overpotential should be expected. The authors need to address the issue further. Voltammetric measurements on pure Al as well as foreign substrate would be helpful to show the characteristics of Al electrodeposition.

2. In third paragraph on page 3, some statements, such as “this interface could be reluctant in having charge transfers, where only part of the Al is willing to accept newly formed Al adatoms” and “as these buds were spherical in shape, Al plating must have occurred at a much slower rate than the ion diffusion inside the electrolyte”, are wrong. In fact, at high overpotential, electron transfer process is accelerated, which could lead to the diffusion of reactant in the electrolyte to be rate-determining step.

3. Figure 2g shows the morphology difference of deposited Al on pure Al and Al-LM anodes, which are bud-like and dendritic, respectively. However, it is understood that Al deposition suffers from dendrite growth, why the dendritic morphology on the Al-LM anode is believed superior to the bud-like shaped one?

4. The role of Ga is elusive. The Al-LM anode with extended treatment has rich boundaries between Al and Ga covered regions which become the nucleation sites and promote fast charging performance. However, the Ga liquid domains transform to spheres or particles after charging. The droplets of the liquid Ga domain are so big after charging, Figure 3, which would significantly reduce boundaries areas and degrade the anode. How can fast-charging performance be maintained in the follow-up cycling?

5. It is rather unusual that the sites at the boundary, rather than on Al, are preferential nucleation sites. Although DFT calculation were finely performed, it did not seem to take the solution as well as electric field influences into considerations. The calculated results may largely deviate from real ones.

6. Despite of lengthy description, the Raman results are not consistent from each other and with the model of dynamic transition in EDL. During charging, EMI⁺ is expected to locate next to the surface so that enhancement of Raman signals from EMI is possible. However, the bands at 753, 790, 1135, 1410, 1590 cm⁻¹, all from EMI, do not appear harmonically at the two Al-LM anodes. The authors hypothesized that the EMI can adopt a variety of molecular configuration as the active anode is donating electrons. On the other hand, the authors hypothesized “if we consider this cationic layer

as the tunneling barrier for electrons, tightly packed EMI will lower the tunneling resistance". But if the latter is valid, EMI configurations would have been restricted. Most contradictorily is the appearance of stronger Raman signals of EMI during discharging, when EMI is supposed to be slightly away from the surface and Raman enhancement is less effective. There are too many speculations which are also contradictory from each other. It is highly suggested that experiments be carefully repeated before conclusion before detailed analysis is made.

7. The view from the reference 4 that ionic liquid exists as neutral cation-anion pairs, with very few ions dissociated in the bulk (< 0.1 wt% in certain types) is of dispute and should be cited with care. Actually, the pairing and dissociation of ions very much depend on water containment in the ionic liquid.

Reviewer #2 (Remarks to the Author):

The manuscript trying to demonstrate the species on anode during operation of Al-ion battery. The issue is important, however there are several concerns in the present manuscript and should be addressed and major-revised before publication.

1. The Raman spectra was acquired at 4 and 8 A g⁻¹, respectively, and the spectra shows different behaviors. It implies that the EDLs is rate-dependent. On the other hand, the battery performance shown in manuscript was measured in 1000 A g⁻¹, 400, 200 and 40 A g⁻¹ rates. The Raman results is representative to explain the behavior at higher rate? How is the battery's performance at 4 and 8 A g⁻¹ charging/discharging rate? Is the performances consistence with observations from the Raman spectra? My opinion is the Raman results seems not well-correlated to battery's performance.
2. The specific capacity of 200 mAh g⁻¹ is unreasonably high. The capacity of Al-carbon battery is mainly resulted from the intercalation of AlCl₄⁻ ions into graphite interlayers. How is the arrangement of chloroaluminate ion can reach such high capacity in graphite layer? The manuscript didn't mention about it and the mechanism is unclear.
3. The author mentioned the peaks at 500 cm⁻¹ (Al₃Cl₁₀⁻) transiently appears and disappears during the charging/discharging cycle. However, from the Raman spectra in Figure 5b, the 500 cm⁻¹ peak starts to appear in charging and remain unchanged during discharging for Al-LM, 8 A g⁻¹ case.
4. Why the Al-triple complex forms only on highly active anode?
5. I am wondering why the author not study the phenomena of active anode in lower rate, such as 100 mA g⁻¹ or 1-C rate.
6. Please address more clearly why supercritical CO₂ in the last drying step can the graphene cathode exhibit smaller redox potentials in CV?
7. The Al-LM alloy surface looks effective in higher rate (>100 A g⁻¹, Figure 2e), but normally the battery not working at this high rate. Please comments on the practical application of Al-LM alloy.
8. In Figure 2e, what is the charging voltage? The inset curve of Figure 2e is not consistent with charging/discharging curve is Supplementary Fig. 3, the capacity of pure Al around 200 mAh g⁻¹ at 100 A g⁻¹ but only ~70 mAh g⁻¹ in Figure 2e. reliability of data is a concern.
9. Please mark the values of normalized capacity in Figure 3c.

Reviewer #3 (Remarks to the Author):

The article “Ultra-fast Charging in Aluminum-Ion Batteries: Electric Double Layers on Active Anode” describes a novel modification of the negative electrode material for an aluminum-graphite battery based on ionic liquid electrolyte. Phase boundaries from the negative electrode is proposed to promote charge transfer with high-flux through the electric double layers. Experimental and theoretical work was combined.

The outstanding features of the work concern:

- specific and successful modification of the negative electrode
- highest reported energy capacity of 200 mAh g⁻¹,
- fastest charging rate of 104 C
- 500% more specific capacity under high rate operations
- explanation and modelling of the underlying chemistry

The article describes a thorough and comprehensive investigation with appropriately utilized experimental and theoretical methods. I can see no shortcomings that prohibit its publication. However, in my opinion, the authors do not rule out the influence of the individual chemical elements of galinstan (gallium, indium, tin) or the residual nickel of the positive electrode on battery performance (ion transport process, modification of the electrolyte, surface effects). Furthermore, the oxidation number of gallium is +3, the same as for aluminum. Therefore, a test of the positive electrode for the intercalation of gallium (or other galinstan species) would be of high interest. This could be realized by an XPS study. It is also not clear what the liquid metal actually does. Does it takes part in the electrochemical process, provide phase boundaries, influence the electric double layer, or has catalytic effect?

In addition, I believe that the article should be paraphrased in order to better focus on the results and their impact on improving aluminium-based batteries. It is not clear what the authors actually want to communicate: a more powerful aluminium-graphite battery (What about energy density and specific energy?) or the influence of the electric double layer (How can the results contribute to the improvement of aluminum-based batteries?) or the synthesis/modification of the negative/positive electrode. For a “communication” it is quite a lot of material. Perhaps the authors can make an effort to better work out the essence of the study and their impact on batteries. The number of figures could be minimized. For example, Fig. 1c does not seem to provide any real information because the scaling is linear and no big difference is obvious, Fig. 2a and Fig. 3a could be combined, Fig. 4 could be improved because in 4a (right) and in 4c (right) the representation appears quite similar.

In my opinion, both the work and the conclusions are original. As the aluminium battery is a promising concept, with high specific energies at cell level expected to benefit from the high aluminium abundance and an already established infrastructure, progress on this battery is of great interest to a broad community (car manufacturers, policy makers, scientists). Looking at both the citation rates and the article views, the topic “aluminium battery” is of increasing interest. As this article describes experimental work, it fills the large gap between theory and application. Therefore, it is topical and of high importance in the field of aluminium-based batteries. In my opinion, however, the addition of gallium to such a battery is of little practical use, as it is one of the rarer elements in the earth’s crust.

The utilized methods are appropriate, and the quality of the data is mainly convincing. Reporting on

data and methodology is mainly sufficiently detailed and transparent to allow for their reproducibility. But, in Fig. 1b, which counter electrode was used? Regarding EIS modelling, neither the program used nor the number of fitted parameters and quality indicators are given. I also miss the convergence estimate of the size of the supercell in the quantum chemical modelling using DFT. Is the used size sufficient for the precision of the calculated adsorption energies? Why were these surfaces used for the modelling? Aluminum is actually polycrystalline, so several surface types should be considered. How high was the lateral resolution of the Raman spectrometer.

The presentation of all data is quite clear and aesthetic. Uncertainties/errors were not given for the evaluated parameters (e.g. resistances: supplementary Fig. 6).

In my opinion, the conclusions and interpretation of the data are robust, valid, and reliable. It would improve the overall representation if the authors were to comment on the influence of the other elements (indium, tin, nickel) on battery performance. This would rule out my concerns about the observed species Al₃Cl₁₀⁻.

I propose the following improvements:

- Paraphrasing the entire text to create a more concise “red thread”, including a description of the impact of the results on the EDL,
- checking/commenting the influence of the other elements (indium, tin, nickel) on the battery performance.

The given references are appropriate. Some statements (see below) should be provided with further references.

The manuscript is clearly written. I would appreciate if the authors could evaluate their results in regard to the improvement for aluminium batteries. What is recommended? While both, the abstract and the section on experimental/theoretical details are clear and appropriate, the “red thread” is not easy to follow and the main text may contain too much detail.

I’m not a specialist in Raman spectroscopy. Therefore, this is outside my scope of expertise. Can the additional elements (gallium, indium and tin) be the third species observed by Raman spectroscopy?

On a more subjective level, I find the article is convincing, very interesting, and well- prepared. Its scientific quality is very high and the comparison with the existing literature is also given. However, the article is sometimes lengthy for a “communication” and does not seem to elaborate sufficiently on the most important points, and the implications for better aluminum-graphite batteries or especially aluminum-ion batteries are not clearly stated and should be covered in a revised version.

The supplementary information is detailed and contributes to a better understanding of the article.

Further comments are:

Page 2, line 47: “Instead of a uniform plating, these adatoms prefer surface defects as nucleation sites.”  Please give a reference.

Page 2, line 57: “In return, this deepens our understanding on EDLs which could shed light on the

future design of other high rate but also high capacity energy storage platforms.”  What does that mean exactly? What follows from the results?

Page 5, line 154: “Such that, a small portion of the anode surface will be modified, with surface pits disappearing first and other areas lightly permeated with gallium.”  Where do these conclusions come from?

Page 5, line 156: “This eventually produces isolated liquid domains that are surrounded by large patches of solid domains.”  Has this been proven?

Page 5, line 168: “we first saw a smooth surface without any pits or cavities.”  A more quantitative discussion of the morphological changes would be helpful.

Page 7, line 260ff: Maybe it is not important to go into detail with all the individual peaks...

Fig. 1: How does a pristine Al anode look like?

Fig. 2g: Are all scale bars the same?

Reviewer #1:

The manuscript by Shen et al. presents a study on the understanding of ultra-fast charging behavior in aluminum-Ion batteries which employ highly active anode (Ga-Al) and 3D grapheme cathode. The ultra-fast charging performance of the Al-ion batteries is impressive and efforts are devoted to elucidate the roles of Ga-Al boundary and electric double layer at the electrode-electrolyte interface, which would be of interest to the large readership of battery and electrochemistry communities. However, there are a number of deficient in the mechanistic aspects of study that makes the work not sound for publication in Nature Comm. Comments are listed as follows.

1. The authors presented two points of views in the introduction section: Pure Al anode is resistive to electron transfer and hinders the production of Al adatoms, and preferential nucleation at defects cast additional barrier for ultra-fast operation, which serve as the origin of problems for the follow-up study of the manuscript. However these views are rather unusual because for Al electrodeposition on pure Al no considerable nucleation overpotential should be expected. The authors need to address the issue further. Voltammetric measurements on pure Al as well as foreign substrate would be helpful to show the characteristics of Al electrodeposition.

We apologize for not making the case clearer. The misunderstanding could come from the electrolyte. State-of-the-art Al-ion batteries are using organic cation-based electrolyte. For example, the electrolyte we prepared (as well as by many others) is by mixing imidazolium chloride (EMI^+Cl^-) (solid) and anhydrous powder of AlCl_3 . An ionic liquid (eutectic mixture) is obtained right after this mixing, by producing three major ions, i.e., Al mono-complex (AlCl_4^-), Al duo-complex (Al_2Cl_7^-), and the organic cation (EMI^+).

Al electrodeposition with this type of electrolyte is not an easy task. Mainly, these negatively charged Al-complexes need to be reduced, by biasing the Al electrode negatively. However, oppositely charged cations (EMI^+) will adsorb on the Al electrode first, by leaving anions as the second adsorption layer. This two-layer structure will then repeat multiple times to form the so-called EDLs. Simply, electrons from the electrode cannot reach those Al-complexes without tunneling through the EMI^+ layer. Besides, reduced $\text{Al}^{(0)}$ adatoms need extra amount of energy before being deposited across the same EMI^+ barrier. Both are the reasons why we say pure Al electrode being resistive in battery charging (or Al electrodeposition). As long as the organic electrolyte is kept unchanged and no other mechanism is provided to overcome the barrier from EMI^+ , this issue in Al electrodeposition will always show up, regardless what type of metals is used as the anode.

As recommended by the reviewer, we selected quite a few candidates (Ag, Ga, In, Sn) and made comparisons with Al and Al-LM. Cyclic voltammogram (**Figure R1**) on the next page showed the details, where 3D graphene is used as the working electrode (cathode), one of those metals above as the reference/counter electrode (anode), and the same eutectic mixture as the organic electrolyte. Except for tin (Sn) that had an irreversible redox reaction, behaviors of all the other electrochemical cells are rather similar. Here we pay attention to two features: location of the major oxidation peak and repeatability of the entire CV scans (multiple scans performed from 0.0 to 2.5 V). The major peak is the place where $\text{Al}^{(0)}$ got electrodeposited on the anode and the 3D graphene cathode was oxidized. Locations for those major peaks varies with different metal anodes, with 2.45 V for In, 2.36 V for Ag, and 2.33 V for Ga. In comparison, Al-LM showed a complete peak for oxidation at a

small potential of 2.35 V. While this number is slightly higher than that for Ga, Al-LM/graphene pair is easier to participate in the redox reactions, with lower oxidation plateaus at 1.55-2.19 V and 2.24-2.42 V but higher reduction plateaus at 1.5-1.92 V and 1.95-2.22 V (see highlights in Fig. R1). Essentially, if we translate these plateaus to performance indicators for batteries, devices using Al-LM/graphene will consume the least amount of energy in charging but release the most energy in discharging. Furthermore, a higher current density for the upper plateau (stronger peaks) indicates that the redox reaction is much more intense. In comparison, although the oxidation potential of gallium is the lowest (2.33 V), its reduction platform is also low and peaks are relatively weak in intensity. Next we compare repeatability of the entire CV scans. It represents how well those electrodeposited Al⁽⁰⁾ can be oxidized back into the organic electrolyte. As expected, pure Al and Al-LM beat all the other candidates in repeatability, in which little difference is observed for multiple CV scans.

Figure R1. Cyclic voltammograms (CV) under a scanning rate of 10 mV s⁻¹. The 3D graphene is the working electrode and Ag/Ga/In/Sn/Al/Al-LM(Ga/In/Sn) as the counter/reference electrode. Major peak around 2.3-2.5 V represents graphene oxidation (accompanied with Al electrodeposition on counter/reference electrode).

We inserted the following paragraphs throughout the manuscript to ease the confusions on Al electrodeposition in an organic electrolyte:

On Page 2, “From chemistry standpoint, metal ions in state-of-the-art Al-ion batteries exist as anionic complexes; the rate of reduction for these large negatively charged ions is much slower than reduction rate of metal salts in water.”

On Pages 3 & 4, “Fast charging at the anode side, however, is not simple. Mainly, Al species inside the organic electrolyte carry negative charges, either as mono-complexed ions (AlCl₄⁻) or duo-complexed ones (Al₂Cl₇⁻). The only way to reduce these Al-complexes is to negatively bias the Al anode. However, this will result in oppositely charged cations (EMI⁺) adsorbing on the anode first, leaving anions no choice but to adsorb as the second layer. Such two-layered structure will then stack on top of one another multiple times to form the so-called EDLs. Due to the presence of EDLs,

electrons from the electrode cannot reach those Al-complexes without tunneling through the EMI⁺ layer. Scanning tunneling microscopy studies in liquid have confirmed such tunneling of electrons through the EMI⁺ barrier. Therefore, the reduced Al⁽⁰⁾ adatoms will need extra amount of energy before being deposited across the same EMI⁺ barrier.”

2. In third paragraph on page 3, some statements, such as “this interface could be reluctant in having charge transfers, where only part of the Al is willing to accept newly formed Al adatoms” and “as these buds were spherical in shape, Al plating must have occurred at a much slower rate than the ion diffusion inside the electrolyte”, are wrong. In fact, at high overpotential, electron transfer process is accelerated, which could lead to the diffusion of reactant in the electrolyte to be rate-determining step.

We appreciate the remarks. While a high overpotential could exhaust the local electrolyte quickly, what we observed in SEM is probably still at the early stage of Al electrodeposition. When a longer duration in charging is conducted, we do expect reactant diffusion being the rate-limiting step. By then, Al growth will become more random in shape (or dendrite-like) (as what we observed in the overcharging experiment in Figure 2f).

To minimize the confusion, we modified the statement on Page 4 as: *“Figure 1h shows that flower buds-like Al grew almost exclusively inside the surface pits. This defect-guided growth suggests a reduction in surface energies being adopted to minimize the total consumption in energy. As those buds were spherical in shape, Al plating must have occurred at the same rate in all directions, implying the ion diffusion inside the electrolyte not yet being used to its extreme (or Al growth being the rate-limiting step)”*.

3. Figure 2g shows the morphology difference of deposited Al on pure Al and Al-LM anodes, which are bud-like and dendritic, respectively. However, it is understood that Al deposition suffers from dendrite growth, why the dendritic morphology on the Al-LM anode is believed superior to the bud-like shaped one?

Both cases mentioned here are overcharging experiments. A battery in “normal” condition will never run under “overcharging” mode. Otherwise, the electrolyte will be exhausted to produce fresh Al layers on anode. This will change the ion compositions in the electrolyte, initiate irreversible side reactions, lower the Coulomb efficiency of the device, and eventually kill the battery.

We performed the overcharging experiment in order to track the Al electrodeposition with an optical microscope. We pay special attention at two fronts: how fast the Al gets deposited and where the deposition takes place. We found: (1) Al electrodeposition on Al-LM is faster; (2) the dendrites on Al-LM are spaced far from each other (perhaps separated by the inactive gallium domain); and (3) a continued Al growth occurs on those dendrites.

Back to superiority, we added a short paragraph on Page 4 to differentiate Al-LM from a pure Al:

“This active anode (Al-LM) is expected to show several advantages. To name a few, the Al growth will no longer be limited to defects. Instead, it will grow over the amorphous boundaries everywhere. Next, each nucleation spot can trigger an explosive growth by forming Al dendrites (Figure 2a-bottom). Large surface areas from the dendrites then promise even higher surface energies for continued Al deposition. As no solid interphase layer will generate from the electrolyte, these dendrites will maintain an intimate contact with Al-LM. Long-term operation of these devices will not be harmed as in lithium or lithium-ion batteries.”

4. *The role of Ga is elusive. The Al-LM anode with extended treatment has rich boundaries between Al and Ga covered regions which become the nucleation sites and promote fast charging performance. However, the Ga liquid domains transform to spheres or particles after charging. The droplets of the liquid Ga domain are so big after charging, Figure 3, which would significantly reduce boundaries areas and degrade the anode. How can fast-charging performance be maintained in the follow-up cycling?*

We suspect the droplets are transformed from those gallium-rich patches, with fewer Al inside. Therefore, they did not reduce the reactive sites that already exist on Al-LM surfaces. As reactive sites are still there, subsequent reactions will still take place. This is the reason we can think of for the uninterrupted cycling behavior.

5. *It is rather unusual that the sites at the boundary, rather than on Al, are preferential nucleation sites. Although DFT calculation were finely performed, it did not seem to take the solution as well as electric field influences into considerations. The calculated results may largely deviate from real ones.*

Let us address this question from three angles: **1)** The main reason Al seems to have lower energy in Ga vicinity is bond-counting effect. Even at (111) surface (both *hcp* and *fcc* sites have 3-fold coordination) Ga atoms seems to adjust to coordinate Al atom more effectively than Al native surface atoms. It reflects a slightly shorter Ga-Al interatomic distance (~ 2.6 Å) allowing Ga atoms near edges to be more accommodating to the presence of Al sterically, i.e. either providing larger number of bonds or allowing binding with second neighbor Al sites to be stronger (due to reduced distance); **2)** We have examined the effect of the external electric field. The average electric field in batteries with well-separated electrodes is not large, \sim V/mm. It does change surface energy due to the charging of metal surface, and strongly affects the motion of molecules in the solvent as the force (qE) per mass of the molecule is very large. However, if atom on the surface of Al does not move significantly the change in potential energy of such atom is small (qEs , where s is a shift due to the external field). We calculated directly the change in the energy of the system as we increase the strength of electric field and we find that at \sim V/mm there very little contribution to the diffusion process, as shown in **Figure R2** (Note: if there are large surface imperfections, say a pyramid, then the diffusion of Al adatom from the base to the top of the pyramid would cause much larger cumulative effect in terms of energy); and **3)** The presence of the ionic liquid should affect the surface processes and we added an appropriate statement in the manuscript (Page 9). The electric double layer formation is expected in charging cycle and, thus, creates sufficiently strong electrostatic interactions as well as lower the mobility of these molecules. Due to the alternating charged layers

Figure R2. Calculation of electric field contribution to the total energy of the Al slab for 2×2 lattice with one Al adatom located either at *fcc* or *hcp* site. DFT calculations were performed in slab geometry, introducing a dipole in the vacuum perpendicular to the plane of the slab. The electric field gives noticeable contribution to the total energy only at values of electric field above ~ 0.01 V/Å. However, it affects both *fcc* and *hcp* locations of adatom nearly the same way. Thus, the diffusion barrier changes very little.

the actual local electric field could be relatively high, ~ 0.01 V/Å. However, on the average the layering should introduce a capacitance like effect and should not change lateral diffusion barriers very significantly. The non-bonding interactions on the average should not contribute differently at different parts of the surface if the EDL is uniform, however, may be important in specific configurations (due to non-uniformities) and may trigger the bonding interactions. This should occur during electrodeposition events.

We investigated the effect of ionic liquid on a bridge-hopping diffusion process for (100) surface. Although this is not the lowest energy event, it should be representative of the change in the electrostatic interactions in surface diffusion. (A concerted motion event is expected to be influenced less by IL). We have included seven $\text{EMI}^+\text{AlCl}_4^-$ complexes placed at Al(100) surface containing a single adatom using 4×4 supercell with 4 Al layers. We performed a DFT relaxation for the lowest energy position of 4-fold coordinated site. Then we fixed molecular position and considered a bridge-hopping event. By keeping the position fixed we overestimating the effect of IL on the diffusion process. The results of DFT calculations in **Figure R3** show that the effect of ionic liquid of the diffusion barrier changes the barrier height from 0.604 to 0.613eV (in weak electrostatic bonding regime). Although during diffusive events the adatom bonding with the ionic liquid molecules changes, the strength of interaction with ionic liquid is order of magnitude smaller than the interaction of adatom with the surface. The loss of some non-bonding pair interaction during diffusion is compensated by formation of new non-bonding pair interactions.

Figure R3. The configuration of a “toy” self-diffusion model of ionic liquid covering Al(100) surface in the lowest energy configuration and a bridge position of Al adatom. Note: we are currently investigating approaches to treat the surface electrochemical reactions. There are multiple obstacles of using DFT-based approaches to treat such events. During such processes the bonding interactions would be introduced and may significantly affect the surface energetics.

6. Despite of lengthy description, the Raman results are not consistent from each other and with the model of dynamic transition in EDL. During charging, EMI^+ is expected to locate next to the surface so that enhancement of Raman signals from EMI is possible. However, the bands at 753, 790, 1135, 1410, 1590 cm^{-1} , all from EMI, do not appear harmonically at the two Al-LM anodes. The authors hypothesized that the EMI can adopt a variety of molecular configuration as the active anode is donating electrons. On the other hand, the authors hypothesized “if we consider this cationic layer as the tunneling barrier for electrons, tightly packed EMI will lower the tunneling resistance”. But if the latter is valid, EMI configurations would have been be restricted. Most contradictorily is the appearance of stronger Raman signals of EMI during discharging, when EMI is supposed to be slightly

away from the surface and Raman enhancement is less effective. There are too many speculations which are also contradictory from each other. It is highly suggested that experiments be carefully repeated before conclusion before detailed analysis is made.

We would like to thank the reviewer for such an excellent point. Perhaps, such behavior of Raman signals indicates more subtle and not so obvious changes in the layers as suggested by our original model.

The new plot in **Figure R4** showing intensity variation with time for $\text{Al}_3\text{Cl}_{10}^-$ and EMI^+ indicates that while in general there is same trend in intensity variations, there are also differences. Bands 790 and 1130 follow changes observed in $\text{Al}_3\text{Cl}_{10}^-$, however, 1410 does not. This might indicate that 790, 1130 and 1410 may belong to different EMI conformers. We hypothesize that certain rearrangements within the EDL may impose rigid structural constraints. For example, planar EMI conformer versus non-planar EMI will show different Raman signatures (computations might reveal more detailed picture). Growth of $\text{Al}^{(0)}$ aluminum or formation of larger intermediate species (Al tri-complex) may significantly alter the regular EDL arrangement, resembling in some way, previously reported pressure induced variations in Raman bands for EMI (Chen, F., et al, *Pressure-induced structural transitions of a room temperature ionic liquid 1-ethyl-3-methylimidazolium chloride. The Journal of Chemical Physics* 2017; Paschoal, V. H., et al, *Vibrational Spectroscopy of Ionic Liquids. Chemical Reviews* 2017). This may serve as another manifestation of dynamic nature of EDL and a variety of possible ionic reconfigurations – that we cannot deduce at the moment due to instrumental limitations.

Figure R4. The intensity variation with time for $\text{Al}_3\text{Cl}_{10}^-$ and EMI^+ under current density of 4A g^{-1} (left) and 8A g^{-1} (right) during 2 cycles of battery operation indicates a coordinated change for both ion species during charging and discharging.

7. The view from the reference 4 that ionic liquid exists as neutral cation-anion pairs, with very few ions dissociated in the bulk ($< 0.1\text{ wt}\%$ in certain types) is of dispute and should be cited with care. Actually, the pairing and dissociation of ions very much depend on water containment in the ionic liquid.

We appreciate the feedbacks from the reviewer. This “reference 4” was selected as it provided a somewhat vivid picture about individual ions in the bulk. However, this reference does not focus on EDLs. As such we followed the suggestion from the Reviewer and removed it from our discussion.

On Page 8, we stated: *“Current research in surface science treats EDLs as stable nanostructures. This includes revealing them as lamellar stacks, interpreting the layered formation with the concept of overcompensation in charge, and capturing nonuniformity over topography defects.”*

Reviewer #2:

The manuscript trying to demonstrate the species on anode during operation of Al-ion battery. The issue is important, however there are several concerns in the present manuscript and should be addressed and major-revised before publication.

1a. The Raman spectra was acquired at 4 and 8 A g⁻¹, respectively, and the spectra shows different behaviors. It implies that the EDLs is rate-dependent. On the other hand, the battery performance shown in manuscript was measured in 1000 A g⁻¹, 400, 200 and 40 A g⁻¹ rates. The Raman results is representative to explain the behavior at higher rate?

We acknowledge the difference in sampling rate. Raman measurements were performed at the highest temporal resolution possible. In particular, spectra were collected at 5-second acquisition time to obtain “nice looking” spectra which allow for resolving signals from different species. Further enhancement in high-rate capture needs an upgrade on facilities (we requested this plus some other modifications in a recent proposal to NSF). Although, the rates are not exactly matched, certain conclusions can still be deduced from the Raman measurements and they include: (1) reaction on anode surfaces can be tracked; (2) new intermediates are discovered with Al-LM being an active anode; (3) new reaction route are proposed to include those intermediates in redox reactions; and (4) the barrier role of EMI⁺ and its conformational change over Al-LM surfaces are explained. These conclusions made it possible to explain how battery performs at higher rates (lowered voltage in charging due to enriched reactive sites from Al-LM), as well as how ions are packed inside the EDLs to facilitate a fast charging.

On Page 10, we added a short paragraph to state the two impacts of Raman measurements: *“It is important to note that, for the new reaction in Eq. 2a to take place, there are two prerequisites. First, the spatial gap between the two duo-complexes (Al₂Cl₇⁻ or AlCl₃·AlCl₄⁻) needs to be small, i.e., less than the van der Waals distance of 5 Å for organic molecules. Such that, a small shift for AlCl₃ from one of the duo-complex to its neighbor can transform the latter anion to a triple-complex (AlCl₃·AlCl₃·AlCl₄⁻). This tight gap further suggests the anionic portion of the EDLs being internally organized more like polymer patches. Inside an individual patch, the Al duo-complex can be regarded as the repeating unit in a conjugated polymer, with much-needed flexibility to reorganize into larger complexes for fast charging. Secondly, extra energy is needed to produce triple-complexes (Al₃Cl₁₀⁻). This explains why we could not capture Al₃Cl₁₀⁻ with pure Al electrode. Mainly, they are only produced around surface defects where surface energy loss compensates for the reaction, but there are just not so many defects available. Al-LM, on the other hand, has plenty of those sites.”*

1b. How is the battery’s performance at 4 and 8 A g⁻¹ charging/discharging rate? Is the performances consistence with observations from the Raman spectra? My opinion is the Raman results seems not well-correlated to battery’s performance.

In comparison to 1000, 400, 200 and 40 A g⁻¹ rates, current densities of 4 and 8 A g⁻¹ are low-rates. In terms of batter’s performance, in **Figure R5**, under the same cut-off voltage (2.45 V) the smaller current density of 4 A g⁻¹ resulted in a lower voltage in charging and a higher capacity. These charge-discharge curves corresponding to Raman spectra in Figure 3f but does not reflect the real limit of

device as saturation voltage for 8 A g^{-1} has not been reached yet. While relatively slow in charging rates, we still managed to capture the changes in EMI^+ signals. Namely, the EMI^+ does not have a fixed conformation. We expect this conclusion being valid under higher rates too. We have requested funds to continue this exploration, by focusing not just on rates but also on individual layers of the EDLs, as well as influences from other ingredients of the electrolyte medium. Again, we appreciate the remarks from the reviewer #2.

2. The specific capacity of 200 mAh g^{-1} is unreasonably high. The capacity of Al-carbon battery is mainly resulted from the intercalation of AlCl_4^- ions into graphite interlayers. How is the arrangement of chloroaluminate ion can reach such high capacity in graphite layer? The manuscript didn't mention about it and the mechanism is unclear.

The charge storage capacity is related to the number of ion adsorbed on the cathode. 3D graphene grown on nickel foam has a large surface area but with few stacked layers. From the XRD on the cathode before and after the charging, not much change in interlayer spacing was observed. We therefore conclude most of the absorptions for chloroaluminate (AlCl_4^-) occurred on open surfaces of graphene. Let us estimate the capacity using a single layer of anions on one graphene monolayer:

We consider the C-C length in graphene with $l = 0.142 \text{ nm}$ and the area of a hexagon is:

$$S_{\text{hexagon-G}} = \frac{3\sqrt{3}l^2}{2} = 5.239 \times 10^{-20} \text{ m}^2$$

In each hexagon, there are 2 carbon atoms ($1/3 \times 6$) so the specific surface area for a single graphene layer (just one side) is:

$$S_G = \frac{S_{\text{hexagon-G}}}{2 * \text{mass of carbon}} = \frac{5.239 \times 10^{-20} \text{ m}^2}{2 \times 1.994 \times 10^{-23} \text{ g}} = 1.314 \times 10^3 \text{ m}^2 \text{ g}^{-1}$$

Next, we take the size of AlCl_4^- as $d = 0.479 \text{ nm}$ (Wang, D. Y., et al. *Advanced rechargeable aluminium ion battery with a high-quality natural graphite cathode*, *Nature Communications* 2017) and assume these Al mono-complexes are closely packed on one-side of a monolayer of graphene. We treat them as a center-filled anionic hexagon, where the area is:

$$S_{\text{hexagon-anion}} = \frac{3\sqrt{3}(d)^2}{2} = 5.961 \times 10^{-19} \text{ m}^2$$

In each hexagon, there will be 3 AlCl_4^- complexes ($1/3 \times 6 + 1$) so the number of close-packed AlCl_4^- per gram of graphene is:

Figure R5. Galvanostatic charge and discharge curves under current densities of 4 and 8 A g^{-1} ($i_c = i_{dc}$). Al-LM is the anode and 3D graphene with a density of 0.16 mg cm^{-2} and mass of 0.03 mg is the cathode.

$$N_{anion} = \frac{3S_G}{S_{hexagon-anion}} = 6.613 \times 10^{21} g^{-1}$$

Theoretical capacity (Q) can be calculated using the Faraday's law, where the number of charge per anion is 1 (for n), F is the Faraday constant, and N_A is the Avogadro's constant:

$$Q_{theoretical} = \frac{nFN_{anion}}{N_A} = \frac{96485.3329 \text{ sA mol}^{-1} \times 6.613 \times 10^{21} g^{-1}}{6.02214 \times 10^{23} mol^{-1}} = 1059.52 \text{ sA g}^{-1} \\ = 294.31 \text{ mAh g}^{-1}$$

Considering that the graphene we made has an open 3D network. Graphene layers are not tightly packed, hence most of absorption will happen on the exposed surfaces. Besides, we did not count the edges from graphene in adsorbing anions. Adding all these factors together, specific capacity can be much greater than 294 mAh g^{-1} . Therefore, our specific capacity of 200 mAh g^{-1} is not unreasonable.

We added above calculations as Note S1 in the Supplementary Materials.

3. The author mentioned the peaks at 500 cm^{-1} ($Al_3Cl_{10}^-$) transiently appears and disappears during the charging/discharging cycle. However, from the Raman spectra in Figure 5b, the 500 cm^{-1} peak starts to appear in charging and remain unchanged during discharging for Al-LM, 8 A g^{-1} case.

We thank the reviewer for pointing this out. The new peak at 500 cm^{-1} ($Al_3Cl_{10}^-$) is indeed transient. We noticed this during the recording of the Raman spectra as well. However, the rate of the peak disappearance does not exactly follow the rate of discharging. Rather, it takes much longer time for these peaks to fully disappear indicating that, perhaps, dynamic changes in EDL structure have certain delay or hysteresis which will be the focus of our future studies.

We added a statement of this transient behavior on Page 7, "It is worthwhile to note that the rate of the peak disappearance does not exactly follow the rate of discharging. Rather, it takes much longer time for these peaks to fully disappear."

4. Why the Al-triple complex forms only on highly active anode?

We addressed this question back in Q#1. Essentially, extra energy is needed to produce triple-complexes ($Al_3Cl_{10}^-$). For pure Al electrode, they are produced around surface defects where surface energy loss compensates for the reaction, but there are just not so many defects available. Al-LM, on the other hand, has plenty of those sites.

5. I am wondering why the author not study the phenomena of active anode in lower rate, such as 100 mA g^{-1} or 1-C rate.

This is a very good question. We used Al-LM as the active anode and performed low-rate charging, as shown in Figure R6. First,

Figure R6. Galvanostatic charge and discharge curves under low current density ($i_c = i_{dc}$). The batteries used Al-LM as the anode and 3D graphene with a density of 0.19 $mg cm^{-2}$ and mass of 0.0303 mg as the cathode.

the saturation voltage to charge the battery varies with the charging current, i.e., a smaller value in voltage plateau for a weaker current. Second, the specific capacities maintained rather constant from 1.0 ~ 20.0 A g⁻¹, but dropped quite a bit when charging current is set at 0.5 A g⁻¹. Mainly, a longer duration in charging will give the battery a chance to electrodeposit Al⁽⁰⁾ over liquid patches of gallium domains. As such, these fresh Al could be dissolved inside the gallium and are not available for subsequent discharging purposes. A reduced Coulomb efficiency in Fig. R6 supports this argument. If we continue to slow down the charging rate, we expect the specific capacity will drop further. In case a balanced performance in both high- and low-rate is needed, we can cut down the infiltration time for Al wire in Galinstan. This will remove much of the redundant liquid patches on surfaces of Al-LM, therefore not sacrificing the low-rate performances. Alternatively, we can do a heat treatment on the Al-LM first, making the liquid patches saturated with Al atoms. Such that, further electrodeposition will leave fresh Al⁽⁰⁾ on surfaces.

6. Please address more clearly why supercritical CO₂ in the last drying step can the graphene cathode exhibit smaller redox potentials in CV?

Supercritically dried graphene will have many open surfaces than regular graphene dried with ethanol during a natural evaporation. In other words, this 3D graphene network will have a much larger surface energy. As a result, covering these fresh surfaces with foreign anions will be much easier. Let us use **Figure R7** as another example. This graph is selected from the reference (*Valota, A. T. et al. Electrochemical Behavior of Monolayer and Bilayer Graphene. ACS Nano 2011*). The different curves indicated that fewer layers do have smaller potentials in oxidation. Overall, comparing to the graphene dried with ethanol in ambient conditions, supercritical CO₂ in the last drying step allows a production of a 3D graphene network with minimum collapsing in structures. This high-surface-energy structure is intrinsically easier to be oxidized by adsorbing negatively charged anions.

Figure R7. Current (normalized to electrode radius) vs. potential response for graphene monolayer (sample 1 and 2), a bilayer, and a multilayer.

7. The Al-LM alloy surface looks effective in higher rate (>100 A g⁻¹, Figure 2e), but normally the battery not working at this high rate. Please comments on the practical application of Al-LM alloy.

We expect devices with Al-LM as the anode eliminates the gap between a supercapacitor and a battery. Therefore, devices with other novel cathodes can all be used to quickly store energy when powerline dropping is expected in a fixed schedule or unexpected with a short notice. This includes energy backup for electric buses that are running between stations, restart a suddenly stopped elevator, or even to minimize power-off-induced loss in manufacturing or production lines.

We added above statement on Page 10-11: “We expect devices with Al-LM as the anode eliminates the gap between a supercapacitor and a battery. Therefore, devices with other novel cathodes can all be used to quickly store energy when powerline dropping is expected in a fixed schedule or unexpected with a short notice. This includes energy backup for electric buses that are running between stations,

restart a suddenly stopped elevator, or even to minimize power-off-induced loss in manufacturing or production lines.”

8. In Figure 2e, what is the charging voltage? The inset curve of Figure 2e is not consistent with charging/discharging curve in Supplementary Fig. 3, the capacity of pure Al around 200 mAh g⁻¹ at 100 A g⁻¹ but only ~70 mAh g⁻¹ in Figure 2e. reliability of data is a concern.

As we mentioned in Supplementary Fig. 4., “the specific capacity of a device is affected by two factors: one is the adsorption and desorption of anions from the graphene and the other is the current density on Al anode. The first one becomes more difficult with the increase of carbon density and the second one becomes larger as carbon mass increases. The latter will contribute to an elevated surface resistance, making charge transfer less efficient (smaller capacity).”

We plotted the range of capacity, showing in Figure 1c, as a red block with capacities ranged from low to high. The high-border for this block includes a group of record-high capacities (200 mAh g⁻¹). To investigate interface reactions between the anode and electrolyte under an injection of a high current density, we no longer limited ourselves to 3D graphene with a small density/mass. This explains why Figure 2e (now changed to Figure 2d) is not consistent with the curve in Supplementary Fig. 3 (now changed to Figure 1d). We added details to the caption of all figures, where the comparison of capacities is involved, as one way to avoid confusions. The voltage information is also added in the caption of Figure 2d as “.... Same cut-off voltage for both cases, saturation voltage of Al-LM anode.”

9. Please mark the values of normalized capacity in Figure 3c.

The purpose of Figure 3c (now changed to Figure 3b) is to depict the dependence of specific capacity on treatment time. The value for specific capacity depends on what mass of graphene is used, not necessarily a constant number. As we mentioned in Question #8, we have many samples with different capacities depending on the mass of cathode. We normalized the capacity to avoid confusions and to emphasize the influence from different treatment times only.

Reviewer #3:

The article “Ultra-fast Charging in Aluminum-Ion Batteries: Electric Double Layers on Active Anode” describes a novel modification of the negative electrode material for an aluminum-graphite battery based on ionic liquid electrolyte. Phase boundaries from the negative electrode is proposed to promote charge transfer with high-flux through the electric double layers. Experimental and theoretical work was combined.

The outstanding features of the work concern:

- specific and successful modification of the negative electrode
- highest reported energy capacity of 200 mAh g⁻¹,
- fastest charging rate of 10⁴ C
- 500% more specific capacity under high rate operations
- explanation and modelling of the underlying chemistry

1. The article describes a thorough and comprehensive investigation with appropriately utilized experimental and theoretical methods. I can see no shortcomings that prohibit its publication. However, in my opinion, the authors do not rule out the influence of the individual chemical elements of Galinstan (gallium, indium, tin) or the residual nickel of the positive electrode on battery performance (ion transport process, modification of the electrolyte, surface effects). Furthermore, the oxidation number of gallium is +3, the same as for aluminum. Therefore, a test of the positive electrode for the intercalation of gallium (or other Galinstan species) would be of high interest. This could be realized by an XPS study. It is also not clear what the liquid metal actually does. Does it takes part in the electrochemical process, provide phase boundaries, influence the electric double layer, or has catalytic effect?

Based on suggestions from reviewer #3, we explored the influence of individual metal elements from Galinstan by varying the compositions. Five samples are involved: pure Al, Al treated by pure gallium (Al-Ga), Al treated with eutectic alloy of gallium (75 wt%) and indium (25 wt%) (Al-Ga/Sn), Al treated with eutectic alloy of gallium (85 wt%) and tin (15 wt%) (Al-Ga/In), and Al treated with Galinstan (Al-Ga/Sn/In). Once one of them is used as the anode, we paired it with a 3D-graphene cathode and the organic electrolyte (EMI-Cl : AlCl₃ = 1.5). Graphs of cyclic voltammogram as well as galvanostatic charge/discharge curves are respectively shown in **Figure R8**. Both Al-Ga/Sn and

Figure R8. Cyclic voltammograms (CV) measured with scanning rate of 10 mV s⁻¹, using 3D graphene as the working electrode and different anodes as the counter/reference electrode (left). Galvanostatic charge and discharge curves with different anodes to 2.45 v (middle) and their own saturation voltages (right).

Al-Ga/Sn/In anodes exhibited the lowest value in potential for the major peak at 2.35 V (left panel), but only Al-Ga/Sn/In had the highest capacity (Fig. R8-middle panel) and lowest charging voltage (Fig. R8-right panel). In **Figure R9**, the battery with Al-Ga/In/Sn demonstrated the best performance in high-rate operations (less decline in capacity). Overall, the liquid metal (Galinstan) we reported in the manuscript is indeed the best anode for Al-ion batteries under high rates.

Figure R9. Specific capacities and Coulombic efficiencies of different anodes under saturation voltages. Current densities varied from 20 to 200 A g⁻¹.

Now we will explain why Al-Ga, Al-Ga/Sn, Al-Ga/In are not as good as Al-Ga/In/Sn. When we performed CV on the single metal (Ga, In, Sn, in **Figure R1**), we found that gallium (Ga) had the lowest oxidation plateaus (1.55-2.19 V and 2.2-2.4 V), but accompanied with low reduction plateaus (1.25-1.67 V, 1.85-2.15 V) and weak peaks during discharging. These all suggest that gallium can reduce interfacial resistance, but too much gallium could dissolve freshly deposited Al, discouraging it for subsequent discharging reactions. Additionally, tin (Sn) had signs of irreversible redox reactions, so it plays a negative role in the battery performance. This matches the observation in Fig. R9 on anode of Al-Ga/Sn, which has a low Coulombic efficiency. Except for the lack of stability, indium (In) seems to have no obvious drawbacks. However, it exhibited the highest value in potential (higher than our set voltage of 2.5 V; Fig. R1). While electrochemically tin is not a favored choice, it does bring down the melting point for liquid metal. Such that, it might have helped a better infiltration through boundaries in aluminum. This is supported by the comparison between Al-Ga/In (Ga : In = 75 : 25 wt%) and Al-Ga/In/Sn in Fig. R9.

We also constructed two electrochemical cells to explore possible intercalation of foreign elements like indium and gallium in the carbon cathode. Here, either a pure indium or a pure gallium is used as the anode but a pyrolytic graphite (PG) is used as the cathode (higher density than 3D graphene therefore heavier loading of anionic compounds). **Figure R10** shows the results, where both anodes functioned well with PG in cyclic voltammogram and battery operations. Once the batteries are overcharged for extended period of time (~10 times more charges are stored), we disconnected the black colored PG from electrochemical cells and placed them on quartz plates for subsequent heating at 850 °C for more than two hours. Leftover objects with a translucent color in white are received. We performed SEM and EDS with these white foams but found no gallium and indium. Note: (1) gallium element in Fig. R10B has a sigma value (1.08) that is much greater than the measurement (0.12). We therefore consider there is no gallium in the PG; (2) As the white foam is quite fragile,

we have to spread it as powders on a graphite tape. This triggered the nontrivial measurement of carbon in Fig. R10B.

Figure R10. No noticeable indium and gallium was captured inside a pyrolytic graphite after overcharging the battery to 10 times of its capacity. Eutectic mixture of EMI-Cl and AlCl₃ are used as the organic electrolyte.

2. In addition, I believe that the article should be paraphrased in order to better focus on the results and their impact on improving aluminium-based batteries. It is not clear what the authors actually want to communicate: a more powerful aluminium-graphite battery (What about energy density and specific energy?) or the influence of the electric double layer (How can the results contribute to the improvement of aluminum-based batteries?) or the synthesis/modification of the negative/positive electrode. For a “communication” it is quite a lot of material. Perhaps the authors can make an effort to better work out the essence of the study and their impact on batteries. The number of figures could be minimized. For example, Fig. 1c does not seem to provide any real information because the scaling is linear and no big difference is obvious, Fig. 2a and Fig. 3a could be combined, Fig. 4 could be improved because in 4a (right) and in 4c (right) the representation appears quite similar.

We highly appreciate the advice from Reviewer #3. We want to present the discoveries on the anode side of the Al-ion batteries. Meanwhile, we want to couple the innovation in anode with reactions at the interface (between this anode and the organic electrolyte). We apologize for not making the manuscript clearer. We have rewritten most of the introduction, rearranged the discussions (especially on Raman), and reduced the number of figures from 5 to 4. We also double checked all the figures (Fig. 1c, Fig. 2a, Fig. 3a, and Fig. 4) to make sure they followed the suggestions from Reviewer #3.

3. In my opinion, both the work and the conclusions are original. As the aluminium battery is a promising concept, with high specific energies at cell level expected to benefit from the high aluminium abundance and an already established infrastructure, progress on this battery is of great interest to a broad community (car manufacturers, policy makers, scientists). Looking at both the citation rates and the article views, the topic “aluminium battery” is of increasing interest. As this article describes experimental work, it fills the large gap between theory and application. Therefore, it is topical and of high importance in the field of aluminium-based batteries. In my opinion, however, the addition of gallium to such a battery is of little practical use, as it is one of the rarer elements in the earth’s crust.

We appreciate the remarks here. Gallium played a special role in our device, being a solvent and being a flexible bonding substrate with Al⁽⁰⁾. While it is rare in the earth’s crust (0.0019%; similar to

lithium of 0.0017%), we do not use much for the surface treatment. Besides, gallium can be easily recycled from Al-LM by soaking in dilute acid (HCl).

4. The utilized methods are appropriate, and the quality of the data is mainly convincing. Reporting on data and methodology is mainly sufficiently detailed and transparent to allow for their reproducibility. But, in Fig. 1b, which counter electrode was used?

We used pure Al as the counter/reference electrode (see description in Methods, Page 12).

5. Regarding EIS modelling, neither the program used nor the number of fitted parameters and quality indicators are given.

We apologize for this. Electrochemical impedance spectroscopy (EIS) measurements were performed using a Gamry Interface 1000E potentiostat in two-electrode mode. This instrument has its own software for fitting. Relevant equivalent circuit model and details can be found in **Figure R11** or in Supplementary Fig. 6.

Figure R11. Circuit model and data fitting for EIS. (a) Relevant equivalent circuit model for EIS data; (b) Nyquist plot; (c) Bode plots and (d) Bode-phase angle versus frequency plots.

In brief, the parameter R_s is the electrolyte resistance; constant phase element (CPE) and R_{CT} are the capacitance and charge-transfer resistance, respectively; and W_0 is the Warburg impedance related to the diffusion of ions into the bulk of the electrode. Total of 6 measurements are performed, i.e., four on Al-LM and two on pure Al. All fittings are shown in Fig. R11a and Figure 2e. Representatives for pure Al anode and Al-LM anode were selected to draw Bode plots (Fig. R11c) and Bode-phase

angle versus frequency plots (Fig. R11d). The simulated results (solid lines) fit very well with experimental data (blue and red colored symbols). Resistances for pure Al and Al-LM are calculated with the model ($R_{CT, \text{pure Al}} = 476.6 \pm 29.60$ ohms; $R_{CT, \text{Al-LM}} = 186.5 \pm 17.79$ ohms).

6. I also miss the convergence estimate of the size of the supercell in the quantum chemical modelling using DFT. Is the used size sufficient for the precision of the calculated adsorption energies? Why were these surfaces used for the modelling? Aluminum is actually polycrystalline, so several surface types should be considered.

We have examined the effect of the supercell on the relative stability of adsorption site and the results are shown in **Figure R12**. The 4×4 cells are usually sufficient to reduce the error due to the repetition of adatom on the surface to semiquantitative analysis. Although Al surface calculations could be tricky, 3×3 and 4×4 surfaces should be sufficient (Roland Stumpf and Matthias Scheffler Phys. Rev. B 53, 4958 1996). In fact, we have previously calculated the diffusion of metal adatom on metallic surface and came to the same conclusion (R. F. Sabiryanov, M. I. Larsson, K. J. Cho, W. D. Nix, and B. M. Clemens Phys. Rev. B 67, 125412 2003]).

In Fig. R12, the variation of the diffusion “barrier” varies within 8 meV for lattices of 3×3 to 5×5 . This is a very small barrier indicating an easy diffusion for isolated Al adatom. Notice that although Al has *fcc* structure, the *hcp* adsorption site has a lower energy. Comparing with the difference in energies of Al-Ga island interactions (~ 1 eV) the energy accuracy for the diffusion barrier is sufficient.

In terms of other surface types, we include the results from (100) and (110) which are other high symmetry surfaces for Al. The surface energy of these surfaces makes them to be less favorable to occur. However, the diffusion of Al adatom in the presence of Ga island fully supports our conclusions based on consideration of diffusion on (111) surface. Al position at the island reduces its energy comparing to sites on the planar surface of the same symmetry. In case of (110) and (100) surface it is basically due to the bond counting effect as Al creates more bonds when attaches or adsorbs on Ga island (Figure R13). Ga atoms appear to be ready to adjust to optimal Al-adatom (compared to native Al surface that is more rigid in that sense). In case of (111) surface the bond counting considerations does not work by symmetry if Ga island would not deform, however, the direct simulation show that Ga effectively surrounding Al-adatom providing stronger bonding.

Figure R12. “Barrier” Energy ($E_{fcc} - E_{hcp}$) as function of the in-plane lattice parameter of the Al slab calculated using PBE-GGA PAW potentials in VASP (Gaussian broadening with $\sigma = 0.2$). Red curve is plotted as $1/r^2 + E_b$ to guide the eye (inverse square behavior reflects elastic fields decay).

Figure R13. Adsorption energy as function of the position of Al adatom on Al(100) and Al(110) surfaces (upper and lower panel, respectively) in the presence of 4-atom Ga island.

7. *How high was the lateral resolution of the Raman spectrometer.*

With optical elements we have in the instrument – the lateral resolution (x-y resolution) is 152 nm. Z-resolution is harder to estimate as it is determined by exponential fall off due to surface plasmon effect. Stronger signal comes from the most adjacent layer – but other layers will also contribute to the signal. Our z-resolution is in the order of a few nm. We can speculate as to how surface roughness might contribute to the spectral effects we observe.

8. *In my opinion, the conclusions and interpretation of the data are robust, valid, and reliable. It would improve the overall representation if the authors were to comment on the influence of the other elements (indium, tin, nickel) on battery performance. This would rule out my concerns about the observed species $Al_3Cl_{10}^-$.*

This issue has been addressed back in Question #1. Basically we found the liquid metal (mixture Ga, In, and Sn) lowering the potential in electrodeposition, but also promoting specific capacity in batteries. Nickel, on the other hand, has been mentioned in reference (Lin, M. C. et al. *An ultrafast rechargeable aluminium-ion battery. Nature 2015*). Therefore we paid special attention in removing it after the growth of graphene.

9. *I propose the following improvements:*

- *Paraphrasing the entire text to create a more concise “red thread”, including a description of the impact of the results on the EDL,*
- *checking/commenting the influence of the other elements (indium, tin, nickel) on the battery performance.*

The given references are appropriate. Some statements (see below) should be provided with further references.

We rewrote most of the introduction to bring the “red thread” (barrier in charge transfer with organic electrolyte), rearranged the discussions (especially on Raman), and reduced the number of figures from 5 to 4. Additional experiments and modeling are also performed to elucidate some of the concerns from the reviewer.

10. *The manuscript is clearly written. I would appreciate if the authors could evaluate their results in regard to the improvement for aluminium batteries. What is recommended? While both, the abstract and the section on experimental/theoretical details are clear and appropriate, the “red thread” is not easy to follow and the main text may contain too much detail.*

We modified our conclusion and added the following comments:

“We expect devices with Al-LM as the anode eliminates the gap between a supercapacitor and a battery. Therefore, devices with other novel cathodes can all be used to quickly store energy when powerline dropping is expected in a fixed schedule or unexpected with a short notice. This includes energy backup for electric buses that are running between stations, restart a suddenly stopped elevator, or even to minimize power-off-induced loss in manufacturing or production lines.”

“One area in our future plan is to investigate the Al deposition in the presence of organic electrolyte. Special attention will be given to the proper analysis of electrostatic interactions with non-uniform surfaces, as these features usually show strong non-local character at the interface of ionic liquids and solids. To push the high-rate operation further, it is imperative to evaluate the insertion of metal cations (Al^{3+}) directly in EDLs, as another boost in charging rate. Not only will it replace those inert

organic cations (EMI^+) by skipping the energy request on electron tunneling, it will also add a 3-electron process to the total reductions.”

11. I'm not a specialist in Raman spectroscopy. Therefore, this is outside my scope of expertise. Can the additional elements (gallium, indium and tin) be the third species observed by Raman spectroscopy?

Gallium chloride ($GaCl_4^-$) has specific Raman band at 345 cm^{-1} (SpectraBase 2020). Indium chloride ($InCl_4^-$) has specific Raman band at 321 cm^{-1} (Kloo, L, et al. Spectroscopic characterization of indium(III) chloride and mixed ligand complexes. Spectrochimica Acta Part A 2002). And tin chloride ($SnCl_4$) has specific Raman band at 368 cm^{-1} (Welsh, H. L. et al, Raman lines of second-order intensity in $SnBr_4$, $SnCl_4$, and CCl_4 , J. Chem. Phys. 1948). These bands would have been quite distinguishable from the bands that we observe for aluminum chloride ($AlCl_4^-$) species.

12. On a more subjective level, I find the article is convincing, very interesting, and well- prepared. Its scientific quality is very high and the comparison with the existing literature is also given. However, the article is sometimes lengthy for a “communication” and does not seem to elaborate sufficiently on the most important points, and the implications for better aluminum-graphite batteries or especially aluminum-ion batteries are not clearly stated and should be covered in a revised version.

We have addressed these concerns in Questions #9-10.

13. Page 2, line 47: “Instead of a uniform plating, these adatoms prefer surface defects as nucleation sites.”  Please give a reference.

We added the following reference: “Unertl, W. N. (1996). Physical structure. Amsterdam, New York, Elsevier”.

14. Page 2, line 57: “In return, this deepens our understanding on EDLs which could shed light on the future design of other high rate but also high capacity energy storage platforms.”  What does that mean exactly? What follows from the results?

We removed this sentence and added one paragraph about future work of our plan on Page 11: “One area in our future plan is to investigate the Al deposition in the presence of organic electrolyte. Special attention will be given to the proper analysis of electrostatic interactions with non-uniform surfaces, as these features usually show strong non-local character at the interface of ionic liquids and solids. To push the high-rate operation further, it is imperative to evaluate the insertion of metal cations (Al^{3+}) directly in EDLs, as another boost in charging rate. Not only will it replace those inert organic cations (EMI^+) by skipping the energy request on electron tunneling, it will also add a 3-electron process to the total reductions.”

15. Page 5, line 154: “Such that, a small portion of the anode surface will be modified, with surface pits disappearing first and other areas lightly permeated with gallium.”  Where do these conclusions come from?

We derived this statement by checking SEM images before and after liquid metal treatment. Cavities, pits, or defects can all be found on a bare Al mesh or wire. A simple soaking in liquid metal removed most of them. The SEM image and elemental mapping of gallium

Figure R14. SEM image and elemental mapping of gallium distribution on surface.

distribution on surface can be found either in **Figure R14** or in Figure 3a. The green parts are Al-rich and gallium only exists in grain boundaries (inset). The purple patches are Ga-rich where the gallium is so much that it can be connected into a whole piece, considered as liquid domains. Obviously, the isolate purple domains (Ga-rich) are surround by the green areas (Al-rich).

16. Page 5, line 156: “This eventually produces isolated liquid domains that are surrounded by large patches of solid domains.”  Has this been proven?

Answer for Question #15 addressed this issue.

17. Page 5, line 168: “we first saw a smooth surface without any pits or cavities.”  A more quantitative discussion of the morphological changes would be helpful.

We appreciate the advice from Reviewer #3. Quantitative analysis of pores in fresh Al is plotted in **Figure R15**, which shows that the fresh Al has a rough texture with many holes of different sizes. After the treatment, we barely see anything on the sample from the SEM image (right).

Figure R15. Quantitative analysis of pores in fresh Al in an area of $60 \times 60 \mu\text{m}^2$ (Size of the pore is in diameter) (right), and the SEM images of the fresh Al (inset) and treated Al (left).

18. Page 7, line 260ff: Maybe it is not important to go into detail with all the individual peaks...

We removed this redundant description on Page 7.

19. Fig. 1: How does a pristine Al anode looks like?

Answer for Question #17 above addressed this concern.

20. Fig. 2g: Are all scale bars the same?

We thank the reviewer for pointing this out. The modified scale bars and the caption are shown in **Figure R16** and Figure 2f.

Figure R15. Over-charging of Al-ion batteries with two different anodes (Al vs. Al-LM, $i_c = i_{dc} = 400 \text{ A g}^{-1}$). ① SEM images of full-charging show early morphologies drastically different; and ②-⑤ are optical microscopy images of front- and side-views of plated Al.

Reviewer #1 (Remarks to the Author):

The authors have answered some of my questions, clarified some confusions, and improved manuscript by rewriting most of the introduction. Unfortunately, some fundamental issues still remain, and reconsideration after further major revision is recommended. Some major issues are explained below along with the first round comment-reply number.

1. First, the influence of the electric double layer structure on Al electrodeposition on Al substrate in EMIAICl₄, which is a mechanistic aspect of the charge transfer process, is regarded as intrinsic barrier in charging in the authors' manuscript. (Such an influence is included in the kinetic parameters such as exchange current and transfer coefficient.) It is expected the same mechanism applies to Al electrodeposition on Al-LM in the same electrolyte, meaning the same barrier exists for Al deposition on Al-LM. But in fact Figure 2a shows another story about difference in Al deposition processes on Al and Al-LM, which are defect mediated or boundary preferred, respectively, which are nothing to do with the above mentioned EDL influence in EMIAICl₄. It is confusing whether the barrier that is reduced with liquid metal, page 4, is the barrier introduced by the EDL in EMIAICl₄. The authors need to clarify the confusion and address the barrier issue more appropriately.

Second, the purpose to perform the CVs is to understand the electrodeposition behaviors on Al and Al-LM, as well as on foreign substrates, which are all anodes of the Al-ion battery. However, the authors took 3D graphene, which is the cathode of the Al-ion battery, as the working electrode, and Al, Al-M and those foreign metals as reference/counter electrodes. This configuration would provide no detailed information about Al electrodeposition. The fact that in the given electrolyte Al and Al-M metal are more stable than those foreign metals during CV measurements leads to differences in the repeatability of the oxidation peaks from the 3D graphene using different metals as the counter/reference electrode. However, it is rather unusual and inappropriate to use such a repeatability difference of processes at the cathode to study the process (i.e. detailed Al electrodeposition) at the anode. Given that the main concern of the manuscript is the anodes with EDL influences, the CVs of at least Al and Al-LM as working electrode should be performed and presented to show the contrast of the Al electrodeposition on these two substrates.

2. The judgment of whether the process is diffusion controlled based on the shape of deposit is not rigorous. Although diffusion controlled condition often causes dendritic growth, the lack of dendrite does not necessarily mean that the process is not under diffusion control. Further rephrases of the description is needed.

5. The "bonding-counting" effect is understood as a result of stronger binding energy of Al on Ga than the cohesive energy of Al. This would lead to the well-known underpotential deposition (UPD) of metal onto foreign metal substrate. However, the UPD ceases after one or two layers only, and the follow-up bulk deposition can only occur at potentials negative of its equilibrium potential. So the "bonding-counting" effect does not explain the exact role of Ga (and the boundary site) on the preferential growth of Al.

6. The authors did not answer the question of why EMI signals are stronger during discharging than during charging, which is against the proposed EDL structure in EMIAICl₄.

Reviewer #2 (Remarks to the Author):

The revised manuscript has been greatly improved and addressed the issue raised by the referees. However, the present form still not adequate for publication. One of the highlights of this work is to propose an active Al-LM surface that facilitate a new mechanism for anode reaction. The evidence is all from the real-time Raman measurements, but the Raman data is still quite confusing. The following issues needs to be furthered clarify before its acceptance for publishing in Nature Comm.

1. the Raman peaks at 753 and 790 cm^{-1} as well as C-C/C-N bond stretches (1135, 1410 and 1590 cm^{-1}) appears for Al-LM surface with charging-discharging at 8 A/g. however, those peaks did not appear harmonically for Al-LM surface with charging-discharging at 4 A/g. It seems the reaction mechanism on the Al-LM surface for 8 A/g and 4 A/g rate is different. It is unusual that the reaction mechanism can be altered by simply changing the charging-discharging rate. How about the Raman spectra for the same Al-LM surface if charging-discharging at 20 A/g and 1.0 A/g?
2. the Raman peak at 500 cm^{-1} , corresponding to triple-complex ($\text{Al}_3\text{Cl}_{10}^-$), starts to appear during charging and remain unchanged during discharging for both cases. The author mentioned this is due to delay or hysteresis of dynamic changes in EDL structure. However, acquisition time for each spectrum was 10 sec long and 10 sec is sufficient to allow dynamic changes in EDL structure. In addition, the Supplementary Fig.15 show that the 500 cm^{-1} peak becomes very weak at cycle 2. Why the mechanism for cycle 1 and cycle 2 is different?

Many contradictory statements between Raman spectrum and the mechanism at which the author trying to proposed. I would like to suggest the author do carefully with Raman measurements and acquire reproducible data since the reaction mechanism proposed is new and needs very strong evidence. If the phenomena on active Al-LM surface is reliable, it would be critical breakthrough to AIBs.

Reviewer #3 (Remarks to the Author):

I already evaluated the manuscript in my first reviewer's report. Therefore, I only comment on the revision.

The revised version as well as the reply to the reviewers show that the authors take all comments serious and made a great effort in improving the manuscript. They addressed all comments, answered them in sufficient detail, and improved the manuscript in respect to the readability, the presentation, the discussion, and the impact of the results. The manuscript now is better focused and the "red line" is fully recognizable. Finally, the authors ruled out all my concerns and I can just congratulate on the efforts and the manuscript.

Please check whether the reference "Unertl, W. N. (1996). Physical structure. Amsterdam, New York, Elsevier" was added to the reference list.

Tilmann Leisegang

REVIEWER COMMENTS (2nd Round)

Reviewer #1:

The authors have answered some of my questions, clarified some confusions, and improved manuscript by rewriting most of the introduction. Unfortunately, some fundamental issues still remain, and reconsideration after further major revision is recommend. Some major issues are explained below along with the first round comment-reply number.

1. First, the influence of the electric double layer structure on Al electrodeposition on Al substrate in EMIAICl₄, which is a mechanistic aspect of the charge transfer process, is regarded as intrinsic barrier in charging in the authors' manuscript. (Such an influence is included in the kinetic parameters such as exchange current and transfer coefficient.) It is expected the same mechanism applies to Al electrodeposition on Al-LM in the same electrolyte, meaning the same barrier exists for Al deposition on Al-LM. But in fact Figure 2a shows another story about difference in Al deposition processes on Al and Al-LM, which are defect mediated or boundary preferred, respectively, which are nothing to do with the above mentioned EDL influence in EMIAICl₄. It is confusing whether the barrier that is reduced with liquid metal, page 4, is the barrier introduced by the EDL in EMIAICl₄. The authors needs to clarify the confusion and address the barrier issue more appropriately.

We believe the confusion originates from the fact that we used the word “barrier” to describe both physical barrier due to EMI⁺ and energy barrier for Al deposition. To avoid such confusion, we changed the heading on page 4 from “*Barrier reduction with liquid metal*” to “*Increasing surface energy with liquid metal*”. We also simplified the sentence on page 5 from “*This performance leap confirmed a lowered energy barrier for electron tunneling and Al(0) depositing*” to “*This performance leap confirmed a lowered energy barrier for Al(0) depositing*”. In addition, we changed “*EMI barrier*” to “*EMI layer*” on page 4. Moreover, Figure 2a was not intended to propose a new mechanism for charge tunneling, but rather to highlight a pathway to gain the energy needed for tunneling through the EDLs, i.e., through Al deposition over amorphous Al-LM.

Second, the purpose to perform the CVs is to understand the electrodeposition behaviors on Al and Al-LM, as well as on foreign substrates, which are all anodes of the Al-ion battery. However, the authors took 3D graphene, which is the cathode of the Al-ion battery, as the working electrode, and Al, Al-M and those foreign metals as reference/counter electrodes. This configuration would provide no detailed information about Al electrodeposition. The fact that in the given electrolyte Al and Al-M metal are more stable than those foreign metals during CV measurements leads to differences in the repeatability of the oxidation peaks from the 3D grapheme using different metals as the counter/reference electrode. However, it is rather unusual and inappropriate to use such a repeatability difference of processes at the cathode to study the process (i.e. detailed Al electrodeposition) at the anode. Given that the main concern of the manuscript is the anodes with EDL influences, the CVs of at least Al and Al-LM as working electrode should be performed and presented to show the contrast of the Al electrodeposition on these two substrates.

We provided CV comparisons of full batteries in our last response, to show electrodeposition behaviors on Al, Al-LM, and foreign substrates being different. Since the same 3D graphene cathode was used, the only difference among these devices is the process of aluminum electrodeposition. The significant conclusion is that, using the Al-LM anode consumes the least amount of energy in charging but releases the most energy in discharging with a stronger redox

intensity. This is evidenced by lower oxidation plateaus for cathode (1.55-2.19 V and 2.24-2.42V; at the same time, the reduction of Al anode takes place), higher reduction plateaus (1.5-1.92 V and 1.95-2.22 V; the oxidation of Al anode takes place) and larger current densities under the same potentials, all indicated in Figure R1-left (extracted from the Figure R1 of our last response). The CV scans therefore did reveal the value of Al-LM in Al electrodeposition.

Furthermore, following the reviewer's suggestion, we performed CV scans without using the 3D graphene cathode, shown in Figure R1-right. Four different metals (Ag, pure Al, Al-LM, and Ga)

Figure R1. Cyclic voltammograms (CV) under a scanning rate of 10 mV s^{-1} . (Left) 3D graphene is used as the working electrode with Al/Al-LM as the counter/reference. (Right) four different metals (Ag/Al/Al-LM/Ga) act as the working electrode, to respectively pair with pure Al (counter/reference).

were respectively used as the working electrode, in which pure Al was used as the counter/reference electrode. It is now clear from these measurements that Al-LM exhibits the highest sensitivity to a given potential (especially comparing to a pure Al electrode), where the reduction process started at the lowest potential among all working electrodes.

2. The judgment of whether the process is diffusion controlled based on the shape of deposit is not rigorous. Although diffusion controlled condition often causes dendritic growth, the lack of dendrite does not necessarily mean that the process not being under diffusion control. Further rephrases of the description is needed.

This is an excellent point. We have now rephrased our statement from “*As those buds were spherical in shape, Al plating must have occurred at the same rate in all directions, implying the ion diffusion inside the electrolyte not yet being used to its extreme (or Al growth being the rate limiting step)*” to “*As those buds were spherical in shape, Al plating must have occurred at the same rate in all directions.*”

5. The “bonding-counting” effect is understood as a result of stronger binding energy of Al on Ga than the cohesive energy of Al. This would lead to the well-known underpotential deposition (UPD) a metal onto foreign metal substrate. However, the UPD ceases after one or two layers only, and the follow-up bulk deposition can only occur at potentials negative of its equilibrium potential. So the “bonding-counting” effect does not explain the exact role of Ga (and the boundary site) on the preferential growth of Al.

We discuss on page 4 and Figure 2a the difference Ga brings to the electrodeposition. Based on the SEM analysis the initial deposition of Al occurs on the Al anode nucleating at the extended surface defect sites (inside pits or cracks) creating “flower buds”. The Coulombic efficacy is lower in such cases. The role of excessive Ga was identified to strongly modify the surface morphology making such defects inaccessible for Al growth. However, partial coverage by Ga causes initial nucleation of Al, either on the smaller scale Al defects or at the Al-Ga boundary where Ga atoms are accommodating Al due to the relaxation. Importantly, Al adatom energy is higher on Al covered with Ga at 1 and 2ML coverage. Thus, initial Al growth most probably will not occur on Ga. In this respect, the picture is drastically different from the UPD, where adatom prefers to deposit onto another material (i.e. Ga in our case).

Ga, thus, coats surface pits (which possess large number of low coordinated Al sites) that act as growth sites of low efficacy. By preventing Al electrodeposition into such locations Al forms growth sites between Ga islands with potential initial nucleation at the Ga-Al boundary. Electrodeposition at such sites shows much higher Coulombic efficacy.

6. The authors did not answer the question of why EMI signals are stronger during discharging than during charging, which is against the proposed EDL structure in EMIAICl₄.

We have observed an excellent correlation between increased intensity of EMI signals and intensity of triple-Al complexes (Fig. 4e). Such correlation prompted us to propose a possible new mechanism for charging that whenever the triple-Al complex is formed it also changes EMI’s relative to the surface configuration – forcing EMI molecule to be closer to the surface (lie down). We depicted such changes in Fig. 4d. Close proximity of EMI to the surface results in stronger enhancement of Raman signals for EMI while concurrently reducing tunneling depth due to smaller EMI layer.

During discharging, signals for the Al triple-complex kept increasing (Fig. 4e). This made the EMI signals increasing too. While we did not provide any explanation for this latter trend, there is a possibility that whenever Al-LM expels one Al³⁺ ion during the discharge process (Al⁰ → Al³⁺), the metal ion can directly couple with two Al single-complexes (AlCl₄⁻) to form one triple-complex (process sketched in Fig. R4; see response to Reviewer #2). This way, there will be a need for this intermediate to grab two extra Cl⁻ from a nearby couple of EMI⁺AlCl₄⁻. This could have freed additional EMI⁺ and caused an increase in concentration of them during discharging.

Figure R2. Confocal laser scanning microscopy images of the dendrites growth (A-C) and dissolution (D-F) on Al-LM. A planar device with two electrodes, i.e., graphene as the working electrode and Al-LM as the counter/reference, was constructed (AlCl₃/EMI-Cl as the electrolyte). We first applied a constant current till a potential of 4.9 V was reached to overcharge this battery and then discharged it under the same current.

Another possibility for increased EMI⁺ signals during discharging could come from the influence of dendrite dissolution which modulate Raman signals due to further enhancement at the fine structural features of dendrites. During the high-speed Al growth, the modified electrode (Al-LM) favors the generation of dendrites unlike pure Al. During discharging, these dendrites becomes thinner, resulting in altered curvatures of their branches and trunks. Here we present a gallery of electrode surface snapshots during both charging (A-C) and discharging (D-F), shown in Figure R2. Fine features of the dendrites will contribute to larger levels of Raman enhancements resulting in stronger electric field and, hence, an exceptional sensitivity of Raman detection for a very little amount of molecular/ionic species (EMI⁺ and others).

Much deeper study is required to correlate fine structures of dendrites with Raman signals before we can precisely describe all the mechanistic details (for example which EMI conformations are favored). We also need more evidences on how the formation of triple-Al complex has correlated with the reconfiguration of EMI. This evidence should come from both charging and discharging, should be obtained with varieties of current densities, and by using Al-LM of different surface compositions.

Reviewer #2 (Remarks to the Author):

The revised manuscript has been greatly improved and addressed the issue raised by the referees. However, the present form still not adequate for publication. One of the highlights of this work is to propose an active Al-LM surface that facilitate a new mechanism for anode reaction. The evidence is all from the real-time Raman measurements, but the Raman data is still quite confusing. The following issues needs to be furthered clarify before its acceptance for publishing in Nature Comm.

1. the Raman peaks at 753 and 790 cm^{-1} as well as C-C/C-N bond stretches (1135, 1410 and 1590 cm^{-1}) appears for Al-LM surface with charging-discharging at 8 A/g. however, those peaks did not appear harmonically for Al-LM surface with charging-discharging at 4 A/g. It seems the reaction mechanism on the Al-LM surface for 8 A/g and 4 A/g rate is different. It is unusual that the reaction mechanism can be altered by simply changing the charging-discharging rate. How about the Raman spectra for the same Al-LM surface if charging-discharging at 20 A/g and 1.0 A/g?

We greatly appreciate the reviewer's comments regarding the mechanism which we offered to explain our experimental observations. We agree with the reviewer that the intensity changes of EMI Raman peaks observed during charging/discharging do not follow the conventional mechanism all that well. While there are some ambiguities, we were able to observe a nice correlation between intensity changes for EMI and triple-Al complex (Fig. 4e), which prompted us to hypothesize about reaction mechanisms with the involvement of EDL reconfiguration (Fig. 4d). Additionally, we are now beginning to understand better that replacing pure Al as anode (in a conventional Al-ion battery) to Al-LM changes how metallic surface behaves during charging/discharging. Complex response of the metallic surface results in deposition of aluminum in a form of dendrites during charging and dendrite dissolution during discharging. We have glanced into the surface dendrites' constant formation and disappearance using confocal scanning microscopy (Fig. R2). Furthermore, different magnitudes of current density contribute to significant variation of the shape, size, and distribution of these dendrites. Fine features of the dendrites contribute to larger levels of Raman enhancements resulting in stronger electric field and, hence, an exceptional sensitivity of Raman detection for a very little amount of molecular/ionic species. We are not surprised that different EMI^+ signals under different current injections can be therefore observed. While, it does not suggest differences in the reaction mechanism where the electrons need to tunnel through the EMI^+ before reaching the negatively charged aluminum complexes, the intensity variations do highlight the complexity of the process which involves configuration changes in EMI^+ packing. Fast growth of dendrites at higher current densities may also disrupt the structure of EDLs resulting in various packing configurations of the species. One possible explanation is that the EMI^+ is a planar-like cation, with an alkyl tail sitting on one corner of the plane. Assembly of these asymmetric cations could adopt varieties of packing configurations, especially over very sharp corners of dendrites where the enhancement effect for Raman tends to be the strongest. While we did measure additional Raman spectra under wide range of current injections (from 0.25 to 160 A g^{-1}), the complexity of dendrite growth on Al-LM electrode suggests further in-depth investigations including careful correlation of dendrite structures with Raman peak variations will be necessary which are beyond this manuscript (see our response to Q2 for details). Such studies will require a combined Raman spectroscopy and confocal microscopy for simultaneous observation of fast events at the Al-LM interface. We plan to address these issues in the future when such method becomes available.

2. the Raman peak at 500 cm⁻¹, corresponding to triple-complex (Al₃Cl₁₀⁻), starts to appear during charging and remain unchanged during discharging for both cases. The author mentioned this is due to delay or hysteresis of dynamic changes in EDL structure. However, acquisition time for each spectrum was 10 sec long and 10 sec is sufficient to allow dynamic changes in EDL structure. In addition, the Supplementary Fig.15 show that the 500 cm⁻¹ peak becomes very weak at cycle 2. Why the mechanism for cycle 1 and cycle 2 is different?

Many contradictory statements between Raman spectrum and the mechanism at which the author trying to proposed. I would like to suggest the author do carefully with Raman measurements and acquire reproducible data since the reaction mechanism proposed is new and needs very strong evidence. If the phenomena on active Al-LM surface is reliable, it would be critical breakthrough to AIBs.

Both Q1 and Q2 of the reviewer have prompted us to investigate further the behavior of the interface during charging and discharging parts of the cycle. We have proposed in our manuscript that Al-LM surface allows for the intermediate triple complex to form easier. The data reported in the manuscript shows that we never observed any triple-complex over the pure Al electrode, however, Al-LM produced the signals for triple-complex varying from charging to discharging. Some disappeared rather quickly, but some stayed almost constant in the entire acquisition window of the Raman scanning. We have further created a more active Al-LM electrode by soaking a piece of Al wire in liquid metal beyond the treatment time used in the manuscript and performed Raman measurements over a wide range of current densities (from 0.25 to 160 A g⁻¹) – Fig. R3. This extensively treated Al-LM electrode offered a different perspective on all participating Al-complexes including single (AlCl₄⁻), double (Al₂Cl₇⁻), and triple (Al₃Cl₁₀⁻) complexes. First, Al single-complex dominates under a small current density of 0.25 A g⁻¹, while Al double-complex dominates under a high current density of 160 A/g. Second, higher degree of variability in Raman intensities is observed at the intermediate current densities (2.5 A g⁻¹ and 40 A g⁻¹). This observation further corroborates our data reported in the manuscript but also points to a complex dependence of Raman intensities on current density and the nature of the interface (Al vs Al-LM_{LOW} vs Al-LM_{HIGH}). Additionally, we have observed that Al triple-complex is always formed for the Al-LM_{HIGH} electrode. Triple-complex does no longer disappear completely but varies in intensity, for all the current densities. We therefore postulate a reasonable explanation to account for all these new findings as:

The combined reaction involving the triple-complex would then be

While above equation (Eq. R3) seems rather different from Eq. 2c in our manuscript, where

Reorganizing Eq. 2c slightly can give us Eq. R3. This can be seen in 3 steps,

Then we follow Eq. 2a, where

Figure R3. Raman signals from Al-LM over a wide window of current densities (from 0.25 to 160 A g⁻¹). To obtain spectral information for such a wide range of current densities we sampled spectra at higher rate, utilized high power laser with 647 nm excitation, and used pure Al as working electrode (instead of graphene). These modifications allowed for sufficient amount of current (or current density per gram of graphene) to flow through the Al-LM (counter/reference) while still make it possible to capture interpretable Raman signals. These factors as well as large reflection from the Al-LM electrode resulted in small intensities of the signal over the entire spectral range. We have focused our analysis on the 250-650 cm⁻¹ window where Al complexes are observed by performing fitting each peak with Lorentzian function for clarity.

High-wavenumber-shift was observed for all the peaks with this experimental setup. The following peaks are assigned: 598 cm⁻¹ - EMI⁺, 311 and 350 cm⁻¹ to Al₂Cl₇⁻ and AlCl₄⁻ respectively, and 529 cm⁻¹ to Al₃Cl₁₀⁻. Slightly larger shift for Al₃Cl₁₀⁻ might indicate further degree of polymerization while staying in the range of peaks between 480 and 540 cm⁻¹ typically assigned to Al₃Cl₁₀⁻ (Dymek, C. J. Jun. , et al. "ChemInform Abstract: Spectral Identification of Al₃Cl₁₀⁻ in 1-Methyl-3-ethylimidazolium Chloroaluminate Molten Salt." ChemInform 19.39, 1988). This gallium rich Al-LM (Al-LM_{HIGH}) resulted in the prominent appearance of Al triple-complexes. The peak intensity does follow similar trend as with relatively low gallium content Al-LM (Al-LM_{LOW}; Figure 3 of the manuscript). Although, strengthening and weakening during charging and discharging, this peak never really disappear under the conditions tested for all current densities, indicating its active participation in electrochemical reactions being realistic.

Applying Eq. 2a to Eq. 2c-2 gives us

Eq. 2c-3 is essentially the same as the new explanation proposed in Eq. R3.

Why are we doing this? This is simply because Eq. R3 or Eq. R1/R2 provides an easier way to understand the reaction than the conventional one ($2\text{Al}_2\text{Cl}_7^- + 3e \leftrightarrow \text{Al}^{(0)} + 7\text{AlCl}_4^-$) for several reasons: (a) a clear connection between all complexes (single-, double-, and triple-) is built; (b) the role of organic electrolyte ($\text{EMI}^+\text{AlCl}_4^-$) in the reaction is provided, i.e., it provides Cl^- and frees EMI^+ from the cation-anion couple; and (c) it shows clearly where the Al^{3+} is going, i.e., it inserts between two Al single-complexes and grabs two free Cl^- from the organic electrolyte. To better illustrate the role of all these complexes, we drew a rough sketch below (Fig. R4) to convey all the essences.

Figure R4. A more inclusive role for the Al triple-complex in discharging. This proposed reaction consumes Al^{3+} and Al single-complexes (AlCl_4^-) but generates triple-complex ($\text{Al}_3\text{Cl}_{10}^-$), dual-complex (Al_2Cl_7^-), and frees EMI^+ from the bulk electrolyte ($\text{EMI}^+\text{-AlCl}_4^-$). This entire process is reversible, from right to the left, for charging.

From descriptions above, we can speculate that the Al-triple species could be both short- and long-lived depending on how active the electrode is and what stage the electrode is at (charging vs. discharging). The intensity variation will also depend on reproducible formation of dendrites which as we indicated in our response to Q1 largely contributes to sensitive detection of species. Since the dendrites exhibit a high degree of structural diversity crossing multiple length scales (from nanometer to micrometer), the enhancement factors over cycles of battery operations are not exactly the same. This difference in surface features can be another reason for the observed intensity fluctuations of the Al triple-complex, thus the differences between cycle 1 and cycle 2 in the data presented in the manuscript as the reviewer correctly pointed out.

Even though we want to conclude that the triple-complex shown in Eq. 2a-2c has perhaps more value than we originally thought, given the complexity mentioned above, the complete description of the mechanism depends on many parameters including nature of the electrolyte, charging/discharging rates, and fine structural features of dendrites. Therefore, we would like to refrain from making any definite mechanistic descriptions. Instead, in the manuscript, we will claim with certainty that we do observe triple Al complex when using Al-LM as electrode (in “Results” section) and state that we believe triple-Al complex plays an important role in the overall

conversion pathway. We will then offer (in the “Discussion” section) what we believe could be most likely mechanism under “Possible New Reaction Route”. We will also reiterate one more time that further detailed investigations will be necessary but will require tools currently unattainable to us.

We made several changes in the manuscript to reflect above discussions:

On page 9, we modified “*New reaction route*” to “*Possible new reaction route*”;

On page 10, we added a statement: “*Secondly, fast charging may not be the only route to produce those Al triple-complexes. In particular, the new anode (Al-LM) while providing much needed high current densities also results in more frequent formation of the triple-complex. Specific details of the new anode’s contribution awaits further explorations. This includes a careful tuning of the surface composition on Al-LM and evaluate its influence to Raman signals, especially under varieties of current densities.*”

Reviewer #3 (Remarks to the Author):

I already evaluated the manuscript in my first reviewer's report. Therefore, I only comment on the revision.

The revised version as well as the reply to the reviewers show that the authors take all comments serious and made a great effort in improving the manuscript. They addressed all comments, answered them in sufficient detail, and improved the manuscript in respect to the readability, the presentation, the discussion, and the impact of the results. The manuscript now is better focused and the "red line" is fully recognizable. Finally, the authors ruled out all my concerns and I can just congratulate on the efforts and the manuscript.

Please check whether the reference "Unertl, W. N. (1996). Physical structure. Amsterdam, New York, Elsevier" was added to the reference list.

Tilmann Leisegang

This literature was no longer needed as we went through a major revision for the introduction. We apologize for this confusion!

Reviewer #1 (Remarks to the Author):

The authors responded carefully to all the comments, and supplemented with some figures to assist the explanation and clarification, and the manuscript has been improved. I appreciate that the CVs taking the anodes as working electrodes were performed. It is strongly suggested that Figure R1 (right) be presented as SI as this is a more professional way of doing electrochemistry. However, the explanations on the experimental data and the influencing factors on the ultra-fast charging of the Al-M anodes are still not sufficiently convincing, largely speculative. Nevertheless given the nice performance of the Al ion batteries, the work may still attract large readership of the energy science community. So I recommend it for acceptance.

Reviewer #2 (Remarks to the Author):

The authors had addressed the issues raised by the referee in details. The Response to Referees Letter clarifying much of the concerns and providing additional experiments/data to support their observation. The comments from the referee in the 2nd round reviewing ask for major revision to the manuscript. However, the present manuscript only made slightly changes compared to the previous manuscript, the points clarified in the Response to Referees Letter doesn't reflect in the manuscript.

I would like to suggest the authors revise the manuscript according to the issues and explanation in 2nd round comment-reply and make all the scientific explanation consistent throughout the whole manuscript.

REVIEWER COMMENTS (3rd Round)

Reviewer #1:

The authors responded carefully to all the comments, and supplemented with some figures to assist the explanation and clarification, and the manuscript has been improved. I appreciate that the CVs taking the anodes as working electrodes were performed. It is strongly suggested that Figure R1 (right) be presented as SI as this is a more professional way of doing electrochemistry. However, the explanations on the experimental data and the influencing factors on the ultra-fast charging of the Al-M anodes are still not sufficiently convincing, largely speculative. Nevertheless given the nice performance of the Al ion batteries, the work may still attract large readership of the energy science community. So I recommend it for acceptance.

We selected the right panel from Figure R1 (last response) as Supplementary Fig. 11 (attached below) on p. 13 of the SI.

Supplementary Fig. 11. Cyclic voltammograms (CV) of Ag, Ga, Al and Al-LM without using the 3D graphene cathode. The scanning rate is 10 mV s⁻¹. Four different metals were respectively used as the working electrode, in which pure Al was used as the counter/reference electrode. It is clear from these measurements that Al-LM exhibits the highest sensitivity to a given potential (especially comparing to a pure Al electrode), where the reduction process started at the lowest potential among all working electrodes.

Reviewer #1 also asked a good question in the second round on whether Al deposition adopted the underpotential pathway. This confusion might arise from the broad readers too. We added the following description on p. 9 of the manuscript:

“The energy landscape of the Al diffusion support the nucleation and growth process described above and illustrated in Figure 2a. Ga strongly modifies the surface morphology making native defect sites inaccessible for Al growth (preventing low Coulombic efficiency). Al diffuses away from Ga-covered surface towards the free Al surface and nucleates at the Al-Ga disordered interface of Ga-free surface. Thus, the directed diffusion increases Coulombic efficiency and prevents the passivation of the electrode due to the multilayer coverage (observed, for example, in underpotential deposition conditions²⁴).”

Reviewer #2:

The authors had addressed the issues raised by the referee in details. The Response to Referees Letter clarifying much of the concerns and providing additional experiments/data to support their observation. The comments from the referee in the 2nd round reviewing ask for major revision to the manuscript. However, the present manuscript only made slightly changes compared to the previous manuscript, the points clarified in the Response to Referees Letter doesn't reflect in the manuscript.

I would like to suggest the authors revise the manuscript according to the issues and explanation in 2nd round comment-reply and make all the scientific explanation consistent throughout the whole manuscript.

To comply with the reviewer's request, we have incorporated arguments provided in the Response to Referees Letter in the previous round of revision into the manuscript. The following changes have been made:

- (1) In SI, on p. 13, we added Supplementary Fig. 11 (see previous page);
- (2) In SI, on p. 19, we added further description to the legend of Supplementary Fig. 16. *“A smaller current density here (vs. 8 A g^{-1}) shows a different trend that can be assigned to variability of Raman sensitivity towards surface features on anode (e.g., unevenness and dendrites growth).”*

Supplementary Fig. 17. Confocal laser scanning microscopy images of the dendrites growth and dissolution on Al-LM. A planar device with two electrodes, i.e., graphene as the working electrode and Al-LM as the counter/reference, was constructed ($\text{AlCl}_3/\text{EMI-Cl}$ as the electrolyte). A constant current was first applied till a potential of 4.9 V was reached to overcharge this battery (a-c) and then it was discharged under the same current (d-f). During discharging, the dendrites becomes thinner, resulting in altered curvatures of their branches and trunks. Fine features of the dendrites will contribute to stronger electric field and, hence, larger enhancements factors and as a result an exceptional sensitivity of Raman detection for small amount of molecular/ionic species (EMI^+ and others).

- (3) The following statement was added on p. 10 of the manuscript: “It’s worthwhile to point out that the Raman intensity fluctuations of the Al triple-complex are observed for different charging cycles. Such variation of the sensitivity in Raman detection of species is attributed to the formation of dendrites over the active anode surfaces. High degree of dendrites’ structural diversity crossing multiple length scales (from nanometer to micrometer) could largely contribute to variability of enhancement factors over cycles of battery operation (see detailed discussions in Supplementary fig. 17).” In SI, on p. 20, we added Supplementary Fig. 17 (see above).
- (4) On page 10 of the manuscript, we added the following in-depth discussion to address how frequent the new triple-complex ($\text{Al}_3\text{Cl}_{10}^-$) will show up in current densities beyond 4 or 8 A g^{-1} and what role it plays in discharging.

“Apparently, the capture of Al triple-complex over the interface of electrolyte and the anode has challenged the conventional understanding in Al-ion batteries. One would question how frequently this new intermediate will form in current densities beyond 4 or 8 A g^{-1} and what role it plays in discharging. We have further created a more active Al-LM anode by soaking a piece of Al wire in liquid metal beyond the treatment time used above (Al-LM_{HIGH}: 6 h; Al-LM_{LOW}: 4 h) and performed Raman measurements over a wide range of current densities (from 0.25 to 160 A g^{-1} , see Supplementary fig. 18). Extensively treated Al-LM anode did offer a perspective on all participating Al-complexes including single (AlCl_4^-), double (Al_2Cl_7^-), and triple ($\text{Al}_3\text{Cl}_{10}^-$) complexes. First, Al single-complex dominates under a small current density, while Al double-complex dominates under a high current density. Second, higher degree of variability in Raman intensities is observed at the intermediate current densities. This observation further corroborates the data shown in Figure 3f but also points to a complex dependence of Raman intensities on current density and the nature of the interface (Al vs Al-LM_{LOW} vs Al-LM_{HIGH}). Additionally, we have observed that Al triple-complex is always formed for the Al-LM_{HIGH} electrode. Triple-complex does no longer disappear completely but varies in intensity, for all the current densities. We therefore postulate a reasonable explanation for Al-triple complex to account for all these new observations as:

The combined reaction involving the triple-complex is as following

Note Eq. 3c is the same as Eq. 2d when the latter runs in opposite direction (i.e., discharging). Eqs. 3a & 3b provide a simpler view on discharging reaction than the conventional one (Eq. 1: $\text{Al}^{(0)} + 7\text{AlCl}_4^- - 3e \leftrightarrow 4\text{Al}_2\text{Cl}_7^-$) for several reasons: (a) a clear connection among all complexes (single-, double-, and triple-) is built; (b) the role of organic electrolyte ($\text{EMI}^+\text{AlCl}_4^-$) in the reaction is further clarified, i.e., it provides Cl^- and frees EMI^+ from the cation-anion pair; and (c) it shows clearly where the oxidized Al (Al^{3+}) is going, i.e., it inserts between two Al single-complexes and grabs two free Cl^- from the organic electrolyte. A schematic sketch to illustrate these reactions is provided in Supplementary fig. 19. From descriptions above, we can hypothesize that the Al-triple species could be both short- and long-lived depending on how active the electrode is and what stage the electrode is at (charging vs. discharging).”

In SI, on p. 21-22, we added Supplementary Fig. 18 &19 as shown here:

Supplementary Fig. 18. Raman signals from Al-LM over a wide window of current densities. We sampled spectra with a large variation in current density, utilized high power laser with 647 nm excitation, and used pure Al as working electrode (instead of graphene). These modifications allowed for sufficient amount of current (or current density per gram of graphene) to flow through the Al-LM (counter/reference) while still make it possible to capture interpretable Raman signals. These factors as well as large reflection from the Al-LM electrode resulted in small intensities of the signal over the entire spectral range. We have focused our analysis on the 250-650 cm⁻¹ window where Al complexes are observed by performing fitting each peak with Lorentzian function for clarity.

High-wavenumber-shift was observed for all the peaks with this experimental setup. The following peaks are assigned: 598 cm⁻¹ - EMI⁺, 311 and 350 cm⁻¹ to Al₂Cl₇⁻ and AlCl₄⁻ respectively, and 529 cm⁻¹ to Al₃Cl₁₀⁻. Slightly larger shift for Al₃Cl₁₀⁻ might indicate further degree of polymerization while staying in the range of peaks between 480 and 540 cm⁻¹ typically assigned to Al₃Cl₁₀⁻. This gallium rich Al-LM (Al-LM_{HIGH}) resulted in the prominent appearance of Al triple-complexes. The peak intensity does follow similar trend as with relatively low gallium content Al-LM (Al-LM_{LOW}; Figure 3 of the manuscript). Although, strengthening and weakening during charging and discharging, this peak never really disappears under the conditions tested for all current densities, which further validates our hypothesis indicating active involvement of the Al triple-complex in electrochemical reactions.

Supplementary Fig. 19. A more inclusive role for Al triple-complex in discharging. This proposed reaction consumes Al^{3+} and Al single-complexes (AlCl_4^-) but generates triple-complex ($\text{Al}_3\text{Cl}_{10}^-$), dual-complex (Al_2Cl_7^-), and frees EMI^+ from the bulk electrolyte ($\text{EMI}^+-\text{AlCl}_4^-$). This entire process is reversed during charging, from right to left.